# Electron Transfer through a Natural Oxide Layer on Real Metal Surfaces Occurring during Sliding with Polytetrafluoroethylene: Dependence on Heat of Formation of Metal Oxides

**Yoshihiro Momose**

Department of Materials Science, Ibaraki University, 4-12-1 Nakanarusawa, Hitachi 316-8511, Japan;
y.momose@cpost.plala.or.jp

**Abstract:** Electron emission (EE) from real metal surfaces occurring during sliding contact with a polytetrafluoroethylene (PTFE) rider has been investigated using the thermodynamic data of metal oxides and the X-ray photoelectron spectroscopy (XPS) intensity ratio of oxygen/metal on the surfaces. EE was termed triboelectron emission (TriboEE). Rolled metal sheets of 18 types were used. The metal-oxygen bond energy calculated from the heat of the formation of metal oxide, ($D(M–O)$), was shown to be a key factor in dividing the EE into two routes, the so-called Schottky effect and the tunnel effect, due to the surface oxide layer. The metals in periodic groups 4 (Ti and Zr), 5 (V, Nb, and Ta), and 6 (Mo and W) maintained higher values of $D(M–O)$, while, moving down the groups, the TriboEE intensity increased, being ascribed to the former route. In groups 10 (Ni, Pd, and Pt) and 11 (Cu, Ag, and Au), the $D(M–O)$ values decreased moving down the groups, but the TriboEE intensity increased significantly, which can be attributed to the latter route. Furthermore, with the increase in the electrical conductivity of metals, the TriboEE intensity became remarkably high, while the $D(M–O)$ value fell rapidly and became almost constant. The XPS results showed that the dependence of the $D(M–O)$ and XPS metal core intensity on the O1s intensity and the XPS intensity ratio of the O1s/metal core was different between groups 10 and 11 and groups 4, 5, and 6. It was concluded that, under the electric field caused on the real metal surface by the friction with PTFE, the electron from metals with small $D(M–O)$ values predominantly tunnels the surface oxide layer as a surface barrier, while with large $D(M–O)$ values, the electron passes over the top of the barrier.

**Keywords:** electron emission; sliding; PTFE; surface oxide layer; heat of formation of metal oxides; metal-oxide bond energies; tunnel effect; Schottky effect



## 1. Introduction

The information on the properties of electrons at metal surfaces involved in the interaction with environments has been of much interest in fields such as adhesion, coating, corrosion, and catalysis [1–3]. The electron or charge transfer at real metal surfaces has also been of great importance in fields such as tribology [4,5] and triboelectric energy production methods [6]. In order to characterize real surfaces, we assessed electron emission (EE) occurring during three main processes of friction, sliding metal surfaces with polymer riders, thermal assistance, and photo or optical stimulation, and examined the EE characteristics of real surfaces of metals, semiconductors, and other materials remaining in ambient environments [7–11]. Furthermore, we reported the relationships between the EE from various metal surfaces and the adhesion and chemical reactions [12–14].

Previously, we reported the total number of electrons emitted from commercial rolled metal sheets of 18 kinds during sliding contact with a polytetrafluoroethylene (PTFE) rider for 60 min and its relationship with the X-ray photoelectron spectroscopy (XPS) results [7,8]. In these reports, the EE and the total number of emitted electrons were termed tribostimulated electron emissions or triboelectron emissions (TriboEE), and TriboEE total count or TriboEE intensity. In the present study, these terms will be called triboelectron

emission (TriboEE) and TriboEE intensity, as used in [9]. The TriboEE measurement was conducted using a gas-flow Geiger counter with a counter gas called Q gas, which consists of a mixture of He and about 1% iso-$C_4H_{10}$ (isobutane) at atmospheric pressure. We arranged the TriboEE intensity of the metals in vertical columns (group) and horizontal rows (period) on the periodic table [7,8]. In [7], we showed that, moving down the groups of the table, the TriboEE intensity increased in the order Ti < Zr (group 4), V < Nb < Ta (group 5), Mo << W (group 6), Ni < Pd < Pt (group 10), and Cu $\cong$ Ag << Au (group 11). We explained that TriboEE may be associated with the release of electrons transported to the attached PTFE debris from the metal substrate by a surface electric field occurring during the sliding. After that, we reexamined the relationship of the TriboEE intensity to other data such as the values of work function (WF) or photothreshold and surface potential [8]. The main conclusion we reached was that the increase in TriboEE intensity between two adjoining metals in the same period correlated to a decrease in the WF or phototheshold values of metal samples; however, for the metals in groups 4, 5, and 6, moving down the groups of the periodic table, the increase in TriboEE intensity was related to a decrease in the WF and photothreshold values, but with the metals in groups 10 and 11, this relationship could not be confirmed.

The motivation of this study was to investigate the key factors increasing the TriboEE intensity of the metals in groups 10 and 11 other than WF or photothreshold. The interest in this study was directed from TriboEE toward the mechanism of electron transfer through a natural oxide layer on 18 rolled metal sheets. Therefore, the relationship between the TriboEE and XPS data reported in [7,8] and the heat of formation of metal oxides and electrical conductivity of metals was comprehensively re-examined. Masel [1] gave a list of the values of metal-adsorbate bond strength in the periodic table, termed *D(M–C)*, *D(M–N)*, and *D(M–O)*, as calculated by Benziger [15], where *D(M–C)*, *D(M–N)*, and *D(M–O)* represent metal–carbon, metal–nitrogen, and metal–oxygen bond energies, respectively, and found that there was a general trend that metals to the left of the periodic table bind adsorbates more strongly than those to the right of the periodic table. Regarding *D(M−O)*, it was shown that, moving down groups 8, 9, 10, and 11 in the periodic table, the bond strengths decrease, while moving down groups 4 and 5, the bond strengths increase. Masel [1] noticed that these trends are very significant, and often form the basis for catalysis design. According to Yagyu et al. [16], we recalculated the *D(M–O)* values, as shown later. Regarding the chemical features of the metal oxides, the following is given from chemistry texts. According to Schriver and Atkins [17], most acidic oxides are formed by covalent bonds, while most basic oxides have ionic bonds, and metallic elements form mainly basic oxides. Furthermore, it was shown that the property of metal oxides became acidic $\rightarrow$ amphoteric $\rightarrow$ basic with a decrease in the oxidation number of the metal. As seen in Figure 5.5 of [17], metal oxides in period 4 with an oxidation number of 2 (Ti, V, Fe, Co, and Ni) are basic and those of Cu and Zn are amphoteric. Regarding the acid–base concept, Huheey [18] generalized that acidity is a positive characteristic of a chemical species and basicity is a negative characteristic of a chemical species. Furthermore, in the periodic table, the basicity of oxides tends to increase when moving down the periodic group. In the present study, as the oxidation number of metal oxides adsorbed in the vicinity of the metal-oxide interface is thought to be low, the basicity of the oxides is considered to become strong. Furthermore, in the TriboEE process, electrons originating from a metal substrate are first transferred into the oxides through the metal-oxide interface and then partly remain in the oxide before leaving the surface. Therefore, the negative character of the oxide becomes stronger. Thus, it is crucial to quantitatively characterize the electronic property of the metal oxide/metal surface.

Regarding the tunnel effect, Moore [19] describes that it is involved in many important phenomena such as the electron flow across the two metallic wires covered with a thin insulating layer of oxide when placed in contact or the tunneling of electrons through potential barriers at an electrode surface. In this effect, the electrons easily pass through such a barrier; they do not need to go over the top of the barrier. This was the observation

that prompted the present study. The purpose of the present work is to estimate the property of the barrier by thermodynamic data of the heat of formation of metal oxides, electrical conductivity of metals, and XPS results, and to clarify the relationship with the TriboEE intensity. This paper addresses the following: (1) the intensity of electron emission with rubbing time; (2) the XPS spectra before and after TriboEE measurement; (3) the heat of formation of metal oxides with various oxidation numbers; (4) the dependence of the TriboEE intensity on the $D(M–O)$ values; (5) the dependence of the TriboEE intensity and $D(M–O)$ values on the electrical conductivity of metals; (6) the relationship between the $D(M–O)$ values and the XPS intensities of F1s, O1s, O1s, C1s, and metal core spectra after TriboEE measurement; (7) the relationship between the $D(M–O)$ values and the XPS intensities of F1s, O1s, C1s, and metal core spectra before TriboEE measurement; and (8) the scheme of TriboEE.

## 2. Materials and Methods

The experimental procedure was described elsewhere in detail [7]. The present study deals with 18 kinds of commercial rolled metal sheets: Al, Ti, V, Fe, Co, Ni, Cu, Zn, Zr, Nb, Mo, Pd, Ag, Sn, Ta, W, Pt, and Au. The thickness and purity of the metal sheets were in the ranges of 0.1–0.3 mm and 99.2–99.999%. For the TriboEE measurement, the samples were cut to a size of $29 \times 29$ mm$^2$. The size of the polytetrafluoroethylene (PTFE) rider with an inserted iron wire was 1.5 mm in diameter and 10 mm in length, and its weight was 0.05 g. Prior to use, the metal sheets and the PTFE rider were ultrasonically cleaned in a mixture of acetone (30 mL) and petroleum benzine (30 mL) for 15 min, followed by drying in a vacuum for 15 min. The TriboEE measurement was performed during sliding contact with a PTFE rider for 60 min using a gas-flow Geiger counter with a counter gas called Q gas, which consists of a mixture of He and 1% isobutane, added as a quenching gas. Therefore, the atmosphere where the sliding is performed is He at atmospheric pressure. The PTFE rider was continuously rotated on the metal surface by a magnetic stirrer at a rate of 400 rpm and a temperature of 298 K. A voltage of −94 V (acceleration voltage, AV) was applied to the metal sample with respect to the earthed grid of the counter during the TriboEE measurement. For each metal, three samples were used together with a new PTFE rider. The apparatus and procedure of the TriboEE measurement have been reported in more detail [7,9]. The measurement of XPS spectra of F1s, O1s, C1s, and metal core for all metal samples was done before and after the TriboEE measurement using a Shimadzu ESCA 750 spectrometer (Shimadzu, Kyoto, Japan) with an X-ray source of Mg K$\alpha$ (8 kV and 30 mA). The XPS measurement conditions of the binding energy range and the sensitivity factor for each core spectra are given later. The maximum and minimum values for each XPS spectra were recorded by the ESCAPAC 760 data system attached to the XPS spectrometer. For each spectrum, the difference between the maximum and minimum values was divided by the sensitivity factor. The XPS characteristics were termed XPS intensity of F1s, O1s, C1s, and metal core spectra. It should be noted that, because, in the case of the Pd sample, the O1s spectrum overlapped with the Pd3p3/2 spectrum, the O1s intensity includes the contribution from this metal.

## 3. Results and Discussion

### 3.1. TriboEE and XPS

The TriboEE commenced immediately on starting the rotation of the PTFE rider and was continuously observed during the rubbing for 60 min. On stopping the rotation, the emission ceased. Figures 1–4 show the intensity (counts/min, abbrev. cpm) of the TriboEE during the rubbing time for all metal samples. The charts are arranged by periodic group. The TriboEE features are described in [7]. The median value of TriboEE intensity is also given in the figures. Hereafter, the average values of TriboEE intensity are used as the TriboEE characteristics for each metal, as shown in Table 1.

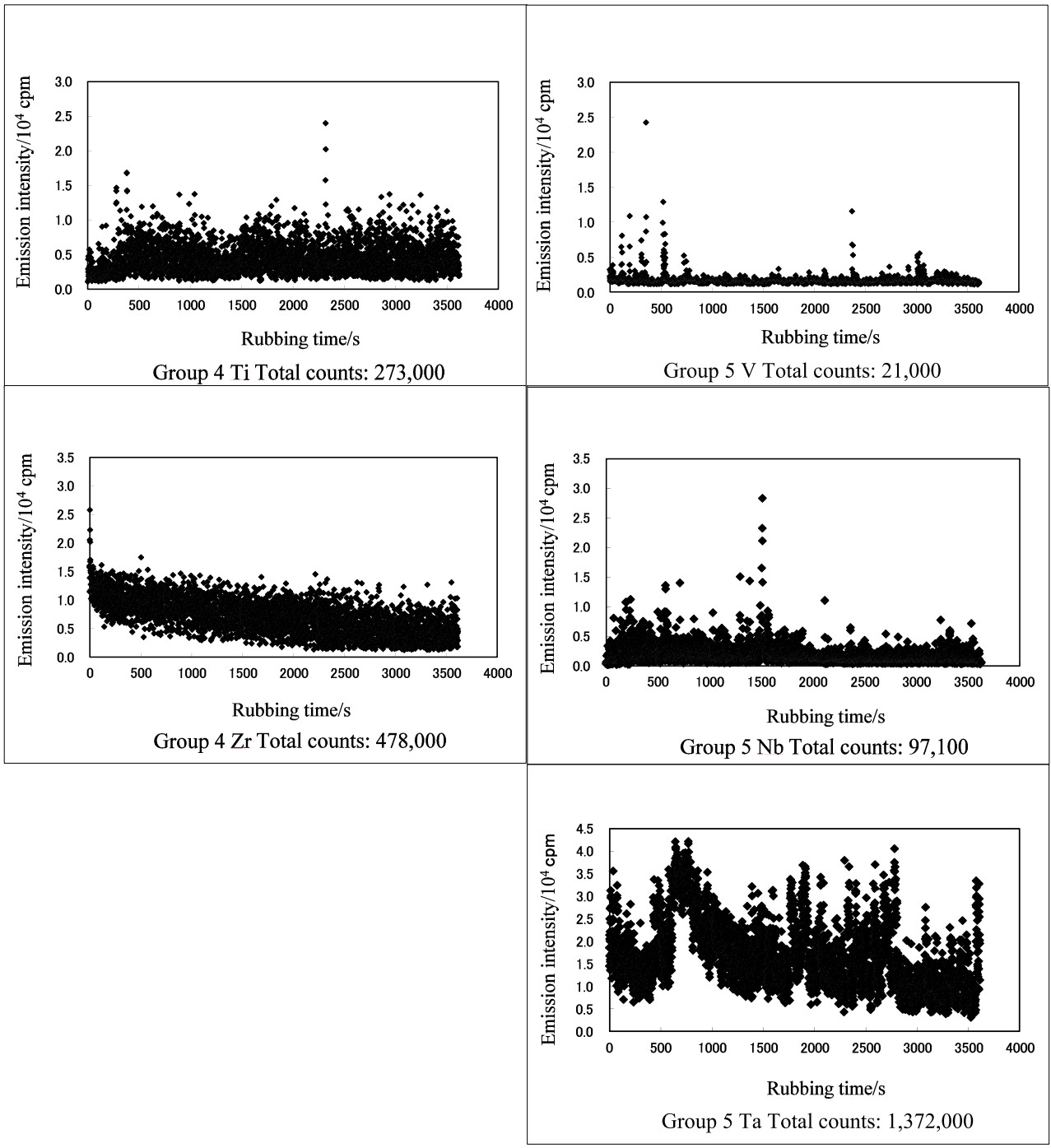

**Figure 1.** The change in intensity (cpm: counts/min) of TriboEE with rubbing time for groups 4 (Ti and Zr) and 5 (V, Nb, and Ta), giving the median value of TriboEE intensity.

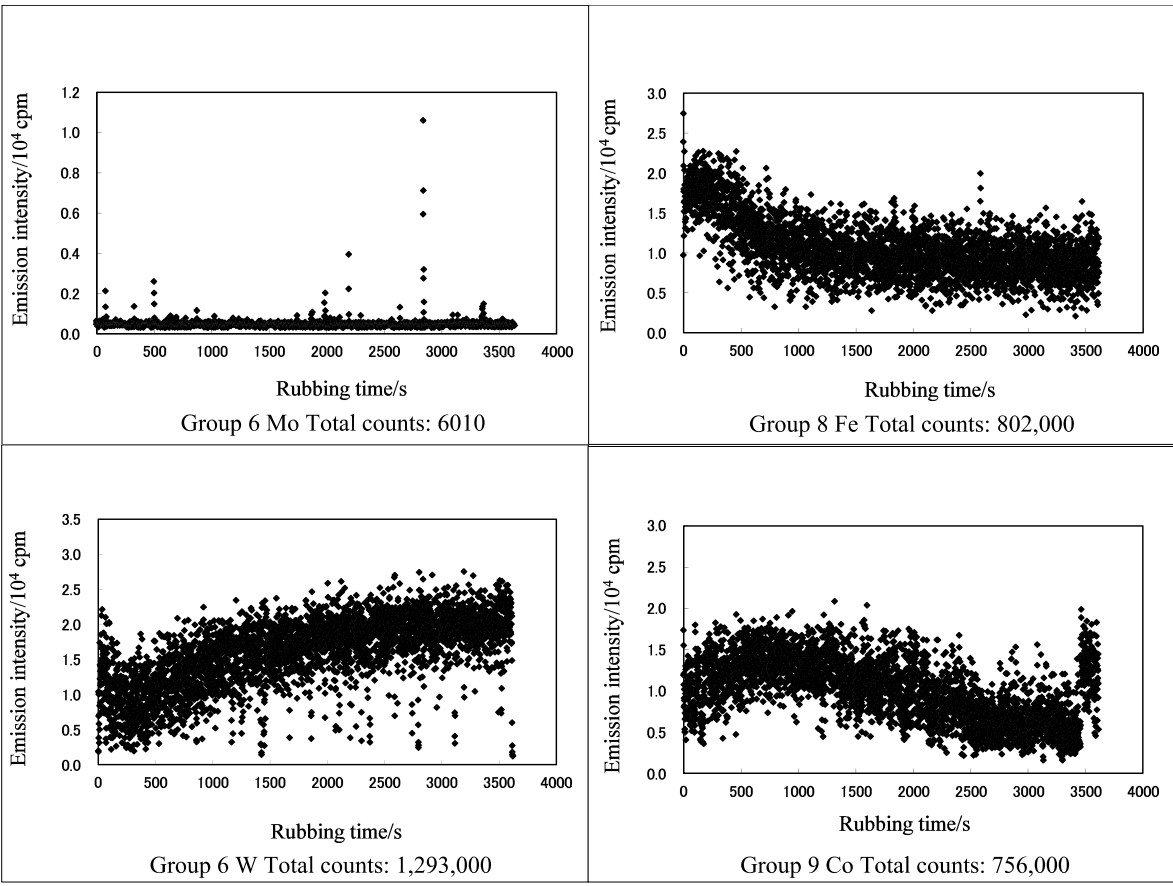

**Figure 2.** The change in intensity (cpm: counts/min) of TriboEE with rubbing time for groups 6 (Mo and W), 8 (Fe), and 9 (Co), giving the median value of TriboEE intensity.

Figures 5–8 show the XPS spectra of F1s, O1s, C1s, and metal core before and after TriboEE measurement for all metal samples, which are arranged by periodic group. The spectra for Au, Ta, W, and Fe samples and the features of the XPS spectra have been previously reported [7]. In the present study, we show the XPS spectra of all metal samples, including the abovementioned metals, in order to thoroughly examine the relationship between the D(M–O) of each metal sample and the O1s and metal core intensities. It should be noted that, for example, the sign of O1s I 10kcps given in the pictures of XPS means that the length of the vertical line on the right side of O1s signal has the X-ray photoelectron emission intensity of 10,000 counts/s. Regarding the O1s spectra shown in Figures 5–8, it should be noted that the oxygen species adsorbed at the metal surfaces mainly originates from a natural oxide layer [7]. The O1s spectra clearly have two components observed at higher and lower binding energies. The former component is the hydroxyl group (-OH), which is observed, for example, at the Au metal surfaces, and the latter is the oxide group ($O_{2-}$), which prevails, for example, at the Ta surfaces. With the Fe samples, components of both hydroxyl and oxide groups appear at the binding energies of 532 eV and 530.5 eV, respectively. For all metal samples, it was found that, with the metals located on the left-hand groups of group 8 (Fe), the adsorption of the oxide group was predominant, while in the metals on the right side of group 8 (Fe), the hydroxyl group was preferentially adsorbed. As the TriboEE intensities for both Au and Ta samples were considerably high (Figures 5 and 7, and Table 1), it was concluded that the TriboEE intensity is poorly correlated with the adsorption modes of the oxygen species. As shown in Table 2, however, moving to the right from group 4 to group 11 in periods 4, 5, and 6, the absolute values of the heat of formation of oxides tend to be lower. The appearance of two oxygen adsorption modes, the oxide and hydroxyl group, may reflect the difference in the heat of formation of oxides.

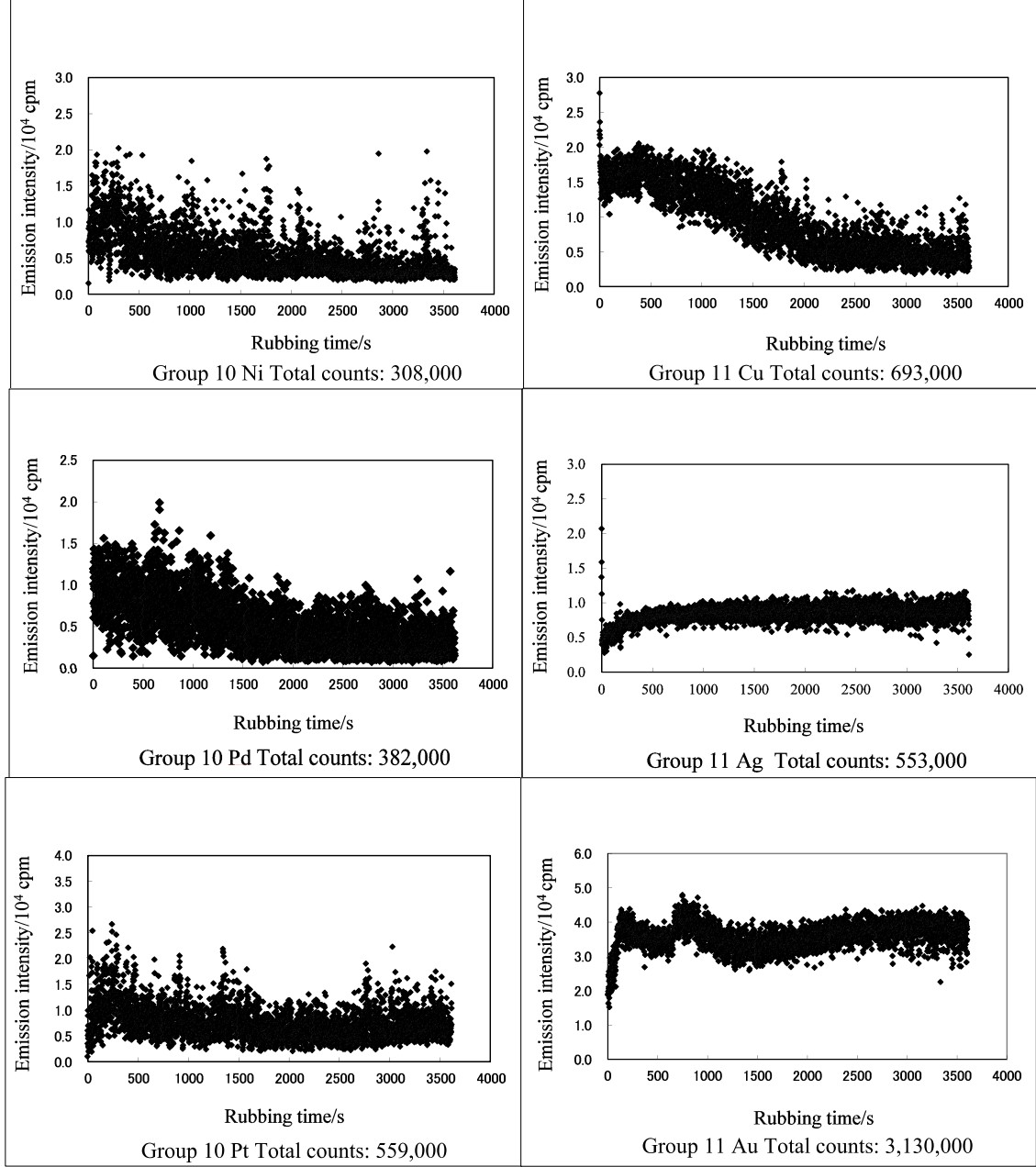

**Figure 3.** The change in intensity (cpm: counts/min) of TriboEE with rubbing time for groups 10 (Ni, Pd, and Pt) and 11 (Cu, Ag, and Au), giving the median value of TriboEE intensity.

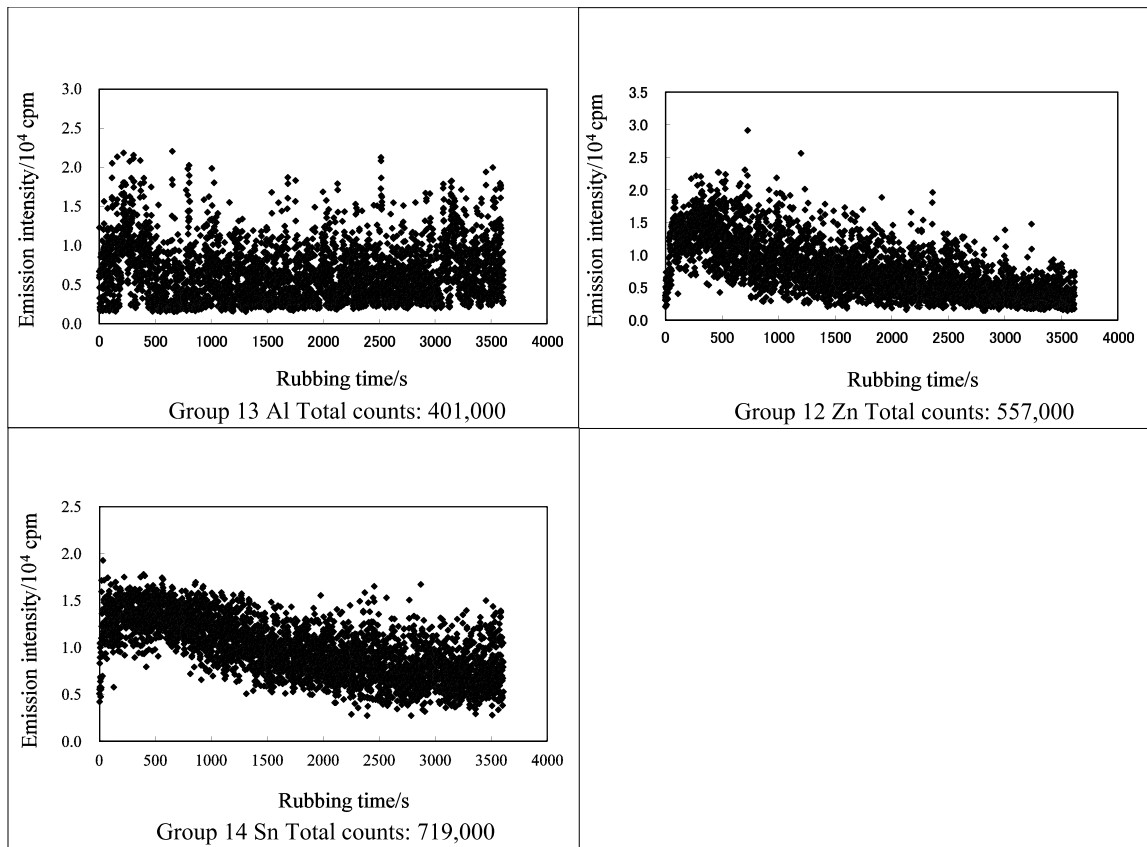

**Figure 4.** The change in intensity (cpm: counts/min) of TriboEE with rubbing time for groups 13 (Al), 12 (Zn), and 14 (Sn), giving the median value of TriboEE intensity.

**Table 1.** The relationship between TriboEE intensity (the average and STDEV values) [1] and the heat of formation of metal oxides per oxygen atom, and the metal–oxygen bond energies, $D(M\text{–}O)$, between the metals and oxygen [2].

| Group | 4 | | 5 | | 6 | | 8 | | 9 | | 10 | | 11 | | 12 | 13 | 14 |
|---|---|---|---|---|---|---|---|---|---|---|---|---|---|---|---|---|---|
| **Element (Period 3)** | - | | - | | - | | - | | - | | - | | - | | - | Al | - |
| TriboEE ($10^5$ counts) Heat of formation of oxide/O (kJ mol$^{-1}$) $D(M\text{–}O)$ ($10^2$ kJ mol$^{-1}$) | - | | - | | - | | - | | - | | - | | - | | - | 3.44 ± 2.66 Al$_2$O$_3$(1/3) −558.5 8.07 | - |
| **Element (Period 4)** | Ti | | V | | - | | Fe | | Co | | Ni | | Cu | | Zn | - | - |
| TriboEE ($10^5$ counts) Heat of formation of oxide/O (kJ mol$^{-1}$) $D(M\text{–}O)$ ($10^2$ kJ mol$^{-1}$) | 2.45 ± 1.22 TiO −519.7 7.68 | > - > | 0.55 ± 0.59 VO −431.8 6.81 | | | | 6.59 ± 2.94 Fe$_3$O$_4$(1/4) −279.6 5.28 | < - > | 7.66 ± 0.86 CoO −237.9 4.87 | > - ≈ | 3.33 ± 1.15 NiO −239.9 4.89 | < - > | 6.83 ± 2.61 Cu$_2$O −168.6 4.18 | > - < | 5.79 ± 1.72 ZnO −350.46 5.99 | - - - | - - - |
| **Element (Period 5)** | Zr | | Nb | | Mo | | - | | - | | Pd | | Ag | | - | - | Sn |
| TriboEE ($10^5$ counts) Heat of formation of oxide/O (kJ mol$^{-1}$) $D(M\text{–}O)$ ($10^2$ kJ mol$^{-1}$) | 5.66 ± 3.87 ZrO$_2$(1/2) −550.3 7.99 | > - > | 1.13 ± 0.86 NbO −405.6 6.54 | > - > | 0.06 ± 0.005 MoO$_2$(1/2) −294.5 5.44 | | - - - | | - - - | | 4.32 ± 1.78 PdO −85.4 3.35 | < - > | 5.03 ± 2.77 Ag$_2$O −31.1 2.80 | | - - - | - - - | 6.29 ± 2.53 SnO$_2$(1/2) −288.8 5.38 |
| **Element (Period 6)** | - | | Ta | | W | | - | | - | | Pt | | Au | | - | - | - |
| TriboEE ($10^5$ counts) Heat of formation of oxide/O (kJ mol$^{-1}$) $D(M\text{–}O)$ ($10^2$ kJ mol$^{-1}$) | - - - | - - - | 11.84 ± 6.32 Ta$_2$O$_5$(1/5) −409.2 6.58 | > - > | 11.14 ± 5.92 WO$_2$(1/2) −294.9 5.44 | | - - - | | - - - | | 5.11 ± 1.26 Pt$_3$O$_4$(1/4) −40.8 2.90 | < - > | 27.25 ± 6.42 Au$_2$O$_3$(1/3) −3.01 2.52 | | - - - | - - - | - - - |

[1] The data come from Momose, Y.; Yamashita, Y. *Tribology International* 2012, *48*, 232–236. [2] The $D(M\text{–}O)$ values are calculated from the largest absolute value in the heat of formation of oxide per oxygen atom for metal oxides, which are given in Table 2, and the heat of formation of oxygen in gas phase, which comes from *CRC Handbook of Chemistry and Physics*, 80th ed.; Lide, D.R. Ed.; CRC Press: Boca Raton, FL, USA, 1999.

**Table 2.** The metal oxides with various oxidation numbers (Roman numerals) in the periods and groups of the periodic table and the values of $\Delta_f H^0$ per oxygen atom of oxides ($\Delta_f H^0$: the heat of formation at 298.15 K (unit: kJ mol$^{-1}$) [1]).

| Group | 4 | 5 | 6 | 8 | 9 | 10 | 11 | 12 | 13 | 14 |
|---|---|---|---|---|---|---|---|---|---|---|
| Oxidation number of metal | - | - | - | - | - | - | - | - | - | - |
| **Period 3** | - | - | - | - | - | - | - | - | **Al** | - |
| III | - | - | - | - | - | - | - | - | Al$_2$O$_3$(1/3) −558.6 | - |
| **Period 4** | **Ti** | **V** | - | **Fe** | **Co** | **Ni** | **Cu** | **Zn** | - | - |
| I | - | - | - | - | - | - | Cu$_2$O −168.6 | - | - | - |
| II | TiO −519.7 | VO −431.8 | - | FeO −272.0 | CoO −237.9 | NiO −239.9 | CuO −157.3 | ZnO −350.46 | - | - |
| II, III | - | - | - | Fe$_3$O$_4$(1/4) −279.6 | - | - | - | - | - | - |
| III | Ti$_2$O$_3$(1/3) −507.2 | V$_2$O$_3$(1/3) −406.2 | | Fe$_2$O$_3$(1/3) −274.7 | - | Ni$_2$O$_3$(1/3) −163.2 | - | - | - | - |
| IV | TiO$_2$(1/2) −472.0 | VO$_2$(1/2) −359.0 | - | - | - | - | - | - | - | - |
| V | - | V$_2$O$_5$(1/5) −310.1 | - | - | - | - | - | - | - | - |
| **Period 5** | **Zr** | **Nb** | **Mo** | - | - | **Pd** | **Ag** | - | - | **Sn** |
| I | - | - | - | - | - | - | Ag$_2$O −31.1 | - | - | - |
| II | - | NbO −405.6 | - | - | - | PdO −85.4 | AgO −11.4 | - | - | SnO −285.9 |
| IV | ZrO$_2$(1/2) −550.3 | NbO$_2$(1/2) −398.1 | MoO$_2$(1/2) −294.5 | - | - | - | - | - | - | SnO$_2$(1/2) −288.8 |
| V | - | Nb$_2$O$_5$(1/5) −379.9 | - | - | - | - | - | - | - | - |
| VI | - | - | MoO$_3$(1/3) −248.4 | - | - | - | - | - | - | - |
| **Period 6** | - | **Ta** | **W** | - | - | **Pt** | **Au** | - | - | - |
| II, IV | - | - | - | - | - | Pt$_3$O$_4$(1/4) −40.8 | - | - | - | - |
| III | - | - | - | - | - | - | Au$_2$O$_3$(1/3) −3.01 | - | - | - |
| IV | - | - | WO$_2$(1/2) −294.9 | - | - | - | - | - | - | - |
| V | - | Ta$_2$O$_5$(1/5) −409.2 | - | - | - | - | - | - | - | - |
| VI | - | - | WO$_3$(1/3) −281.0 | - | - | - | - | - | - | - |

[1] The formula and the values of the heat of formation for oxides are from *CRC Handbook of Chemistry and Physics*, 80th ed.; Lide, D.R. Ed.; CRC Press: Boca Raton, FL, USA, 1999, and *Lange's Handbook of Chemistry*, 12th ed.; Dean, J.A., Ed.; McGraw-Hill, New York, NY, USA, 1979. The heat of formation of oxide for Au comes from *CRC Handbook of Chemistry and Physics*, 62nd.; Weast, R.C., Astle, M.J. Eds.; CRC: Boca Raton, FL, USA, 1981.

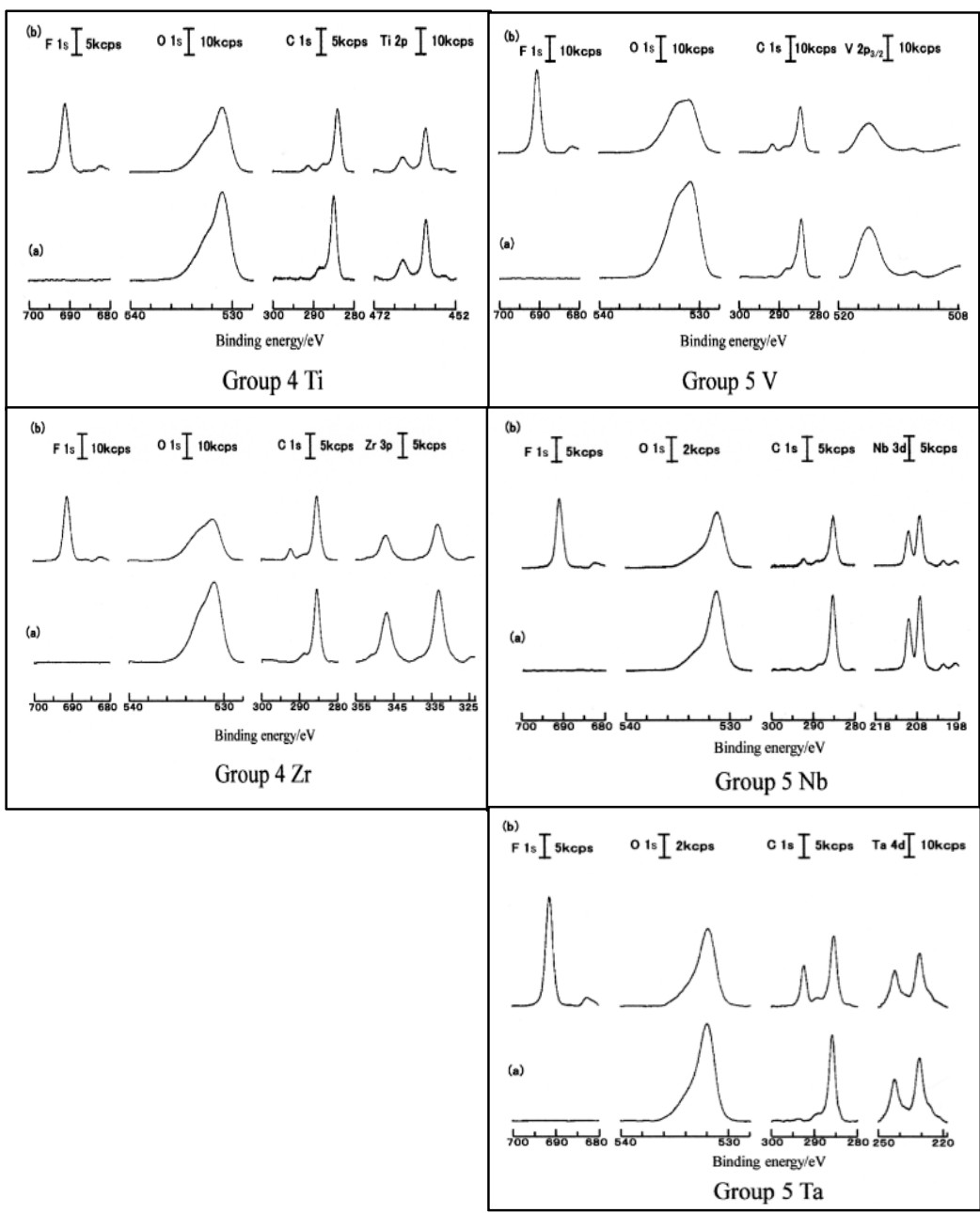

**Figure 5.** X-ray photoelectron spectroscopy (XPS) spectra for groups 4 (Ti and Zr) and 5 (V, Nb, and Ta) (**a**) before and (**b**) after the TriboEE measurement.

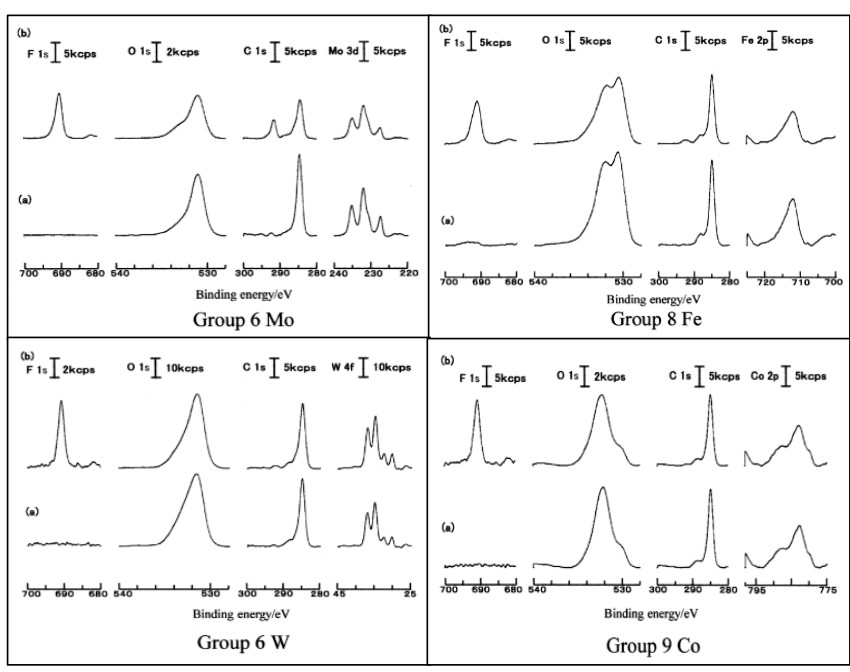

**Figure 6.** XPS spectra for groups 6 (Mo and W), 8 (Fe), and 9 (Co) (**a**) before and (**b**) after the TriboEE measurement.

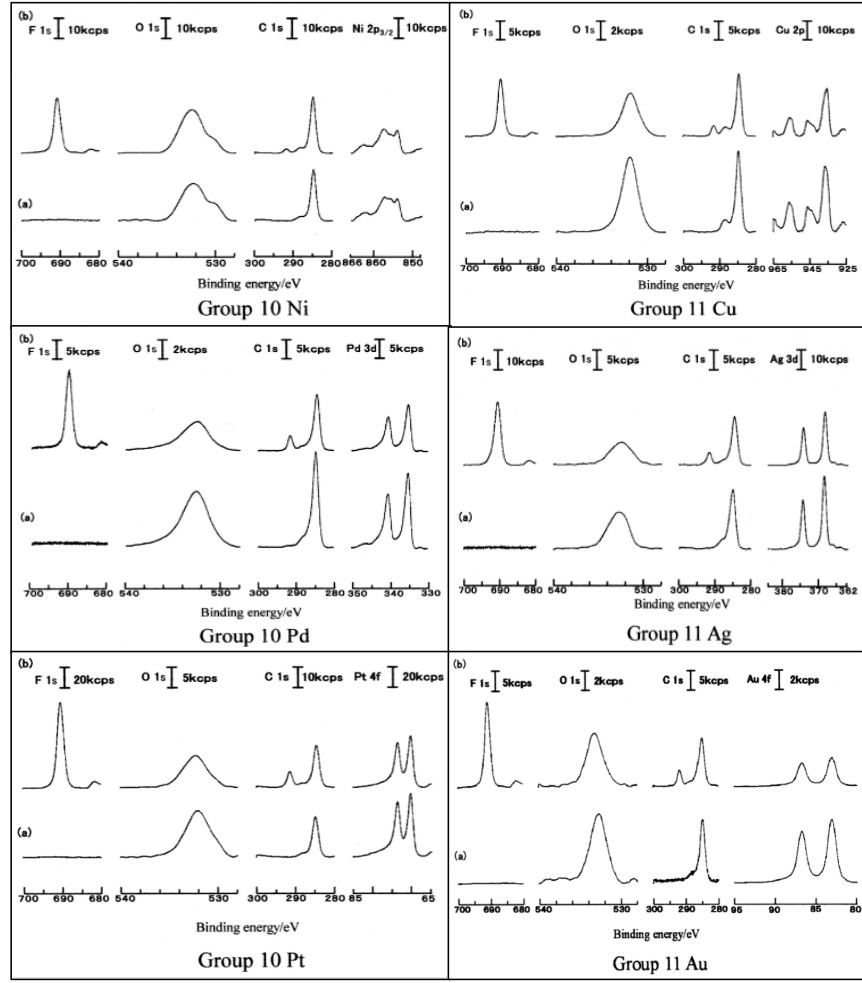

**Figure 7.** XPS spectra for groups 10 (Ni, Pd, and Pt) and 11 (Cu, Ag, and Au) (**a**) before and (**b**) after the TriboEE measurement.

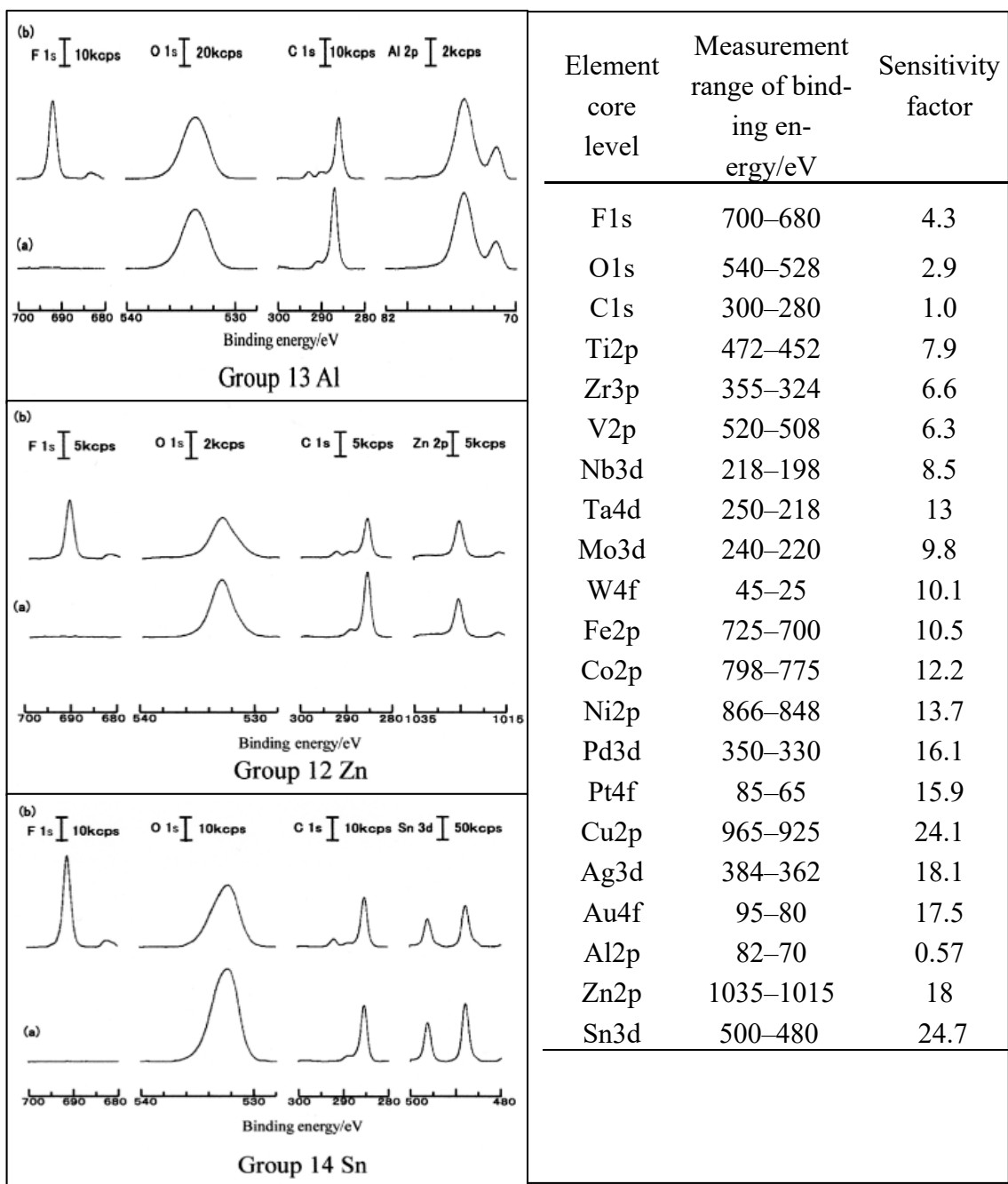

| Element core level | Measurement range of binding energy/eV | Sensitivity factor |
|---|---|---|
| F1s | 700–680 | 4.3 |
| O1s | 540–528 | 2.9 |
| C1s | 300–280 | 1.0 |
| Ti2p | 472–452 | 7.9 |
| Zr3p | 355–324 | 6.6 |
| V2p | 520–508 | 6.3 |
| Nb3d | 218–198 | 8.5 |
| Ta4d | 250–218 | 13 |
| Mo3d | 240–220 | 9.8 |
| W4f | 45–25 | 10.1 |
| Fe2p | 725–700 | 10.5 |
| Co2p | 798–775 | 12.2 |
| Ni2p | 866–848 | 13.7 |
| Pd3d | 350–330 | 16.1 |
| Pt4f | 85–65 | 15.9 |
| Cu2p | 965–925 | 24.1 |
| Ag3d | 384–362 | 18.1 |
| Au4f | 95–80 | 17.5 |
| Al2p | 82–70 | 0.57 |
| Zn2p | 1035–1015 | 18 |
| Sn3d | 500–480 | 24.7 |

**Figure 8.** XPS spectra for groups 13 (Al), 12 (Zn), and 14 (Sn) (**a**) before and (**b**) after the TriboEE measurement and the measurement range of binding energy and sensitivity factor for the core level XPS spectra of F1s, O1s, C1s, and 18 kinds of metals.

### 3.2. Heat of Formation of Metal Oxides with Various Oxidation Numbers

In Table 2, metal oxides with various oxidation numbers and the values of the heat of formation of oxides, $\Delta_f H^0$, (unit: kJ·mol$^{-1}$) per oxygen atom are classified into periodic periods and groups in order of the oxidation number. The $\Delta_f H^0$ values for oxides formula are selected from the *CRC Handbook of Chemistry and Physics* and *Lange's Handbook of Chemistry* [20–22]. The values of $\Delta_f H^0$ per oxygen atom are obtained by dividing the value of $\Delta_f H^0$ by the number of oxygen atoms in the oxides. For example, the values of $\Delta_f H^0$ in the case of iron oxides are −1118.4 for $Fe_3O_4$, −824.2 for $Fe_2O_3$, and −272.0 kJ·mol$^{-1}$ for FeO. The values of $\Delta_f H^0$ per oxygen atom are as follows: $Fe_3O_4$ (1/4): −279.6, $Fe_2O_3$

(1/3): −274.7, and FeO: −272.0 kJ·mol$^{-1}$. For each metal, the values of $\Delta_f H^0$ per oxygen atom of the oxides are classified by the oxidation number. It is seen that, for oxides of each metal, except Fe and Sn, the absolute values of $\Delta_f H^0$ per oxygen atom tend to increase with a decreasing oxidation number. We consider that the natural oxide layer predominantly consists of the oxide with the highest absolute value of $\Delta_f H^0$ per oxygen atom.

From Table 2, the following observations are highlighted:

1.　As one moves down groups 4, 5, 6, 10, and 11 from period 4 to period 6, the highest absolute value of $\Delta H^0$ per oxygen atom in periodic groups 4, 5, and 6 does not change uniformly, as follows: TiO < ZrO$_2$ (1/2) (Group 4); VO > NbO < Ta$_2$O$_5$ (Group 5); WO$_2$ ≈ MoO$_2$ (Group 6), while moving down groups 10 and 11 it smoothly decreases as follows: NiO > PdO > Pt$_3$O$_4$ (1/4) (Group 10); Cu$_2$O > Ag$_2$O > Au$_2$O$_3$ (1/3) (Group 11).

2.　As one progresses to the right from group 4 to group 6 in periods 4 and 5, the absolute value of $\Delta_f H^0$ per oxygen atom with the same oxidation number decreases in the following order: TiO > VO (Period 4); ZrO$_2$ (1/2) > NbO$_2$ (1/2) > MoO$_2$ (1/2) (Period 5); in period 6, although the oxidation number decreases from V to IV, it decreases in the following order: Ta$_2$O$_5$ (1/5) > WO$_2$ (1/2) (period 6).

3.　Moving to the right from group 10 to group 11 in periods 4, 5, and 6, the absolute value of $\Delta_f H^0$ per oxygen atom decreases in the following order: NiO > CuO (period 4); PdO > AgO (period 5); Pt$_3$O$_4$ (1/4) > Au$_2$O$_3$ (1/3) (period 6).

The orders of the absolute value of $\Delta_f H^0$ per oxygen atom in the above correspond to those of the TriboEE intensities in Table 1.

### 3.3. TriboEE Intensity and D(M−O)

According to Benziger [15], who proposed that the adsorption enthalpy $\Delta H_{ad}$ (*A*) for *A* adatom on the metal surface is approximately equal to the normalized enthalpy of bulk compound $(1/y)\Delta H_f·(M_x A_y)$, Yagyu et al. [16] obtained the *D(M–O)* values for metal oxides using Equation (1):

$$D(M\text{-}O) = -(1/y)\, \Delta_f H^0\, (M_x O_y) + \Delta_f H^0\, (O(g)) \tag{1}$$

where *D(M−O)* is the metal−oxygen bond energy; $(1/y)\, \Delta_f H^0(M_x O_y)$ is the normalized heat of formation of metal oxides per oxygen atom; and $\Delta_f H^0(O(g))$ is the heat of formation of oxygen in gas phase, equal to +249.2 kJ mol$^{-1}$ [20]. For example, let us obtain the iron−oxygen bond energy, *D(Fe−O)*, from the data of the heat of formation for Fe$_3$O$_4$ (s) and O (g). These chemical reactions for the enthalpy changes are represented as (a) and (b):

(a)　3Fe(s) + 2O$_2$(g) → Fe$_3$O$_4$(s) $\Delta_f H^0(Fe_3 O_4)) = -1118.4$ kJ·mol$^{-1}$

(b)　(1/2)O$_2$(g) → O(g) $\Delta_f H^0(O(g))= +249.2$ kJ·mol$^{-1}$

The reactions (a) and (b) are rewritten to the thermochemical Equations (c) and (d):

(c)　(1/4) [3Fe(s) + 2O$_2$(g)] → (1/4)[Fe$_3$O$_4$(s) + 1118.4 kJ·mol$^{-1}$]

(d)　(1/2)O$_2$(g) → O(g) − 249.2 kJ mol$^{-1}$

Note that (c) stands for the formation of Fe$_3$O$_4$(s) per oxygen atom: (1/4)Fe$_3$O$_4$ from (3/4)Fe(s) and (1/2)O$_2$(g). The reaction (c)−(d) becomes (e). In this case, the heat of 528 kJ·mol$^{-1}$ is produced as given in (e),

(e)　(3/4)Fe(s) + O(g) →(1/4)Fe$_3$O$_4$(s) + [(1/4) × 1118.4] kJ·mol$^{-1}$ + 249.2kJ·mol$^{-1}$.

Thus, *D(Fe−O)* = 528 kJ·mol$^{-1}$ (Table 1).

From the values of $\Delta_f H^0(M_x O_y)$ per oxygen atom given in Table 2, the *D(M–O)* values were determined from the absolute highest value of $\Delta_f H^0(M_x O_y)$ per oxygen atom for each metal in the same way as in [16]. The *D(M–O)* value of Au was obtained from the heat of formation of Au$_2$O$_3$ of −9.04 kJ mol$^{-1}$ [22]. In Table 1, the TriboEE intensity (the average and STDEV), the heat of formation of oxide per oxygen atom, and *D(M–O)* are listed.

Figures 9 and 10 show the relationship between the TriboEE intensity and the *D(M–O)* values. From Figure 9, we can draw the following conclusions. (1) In groups 4 (Ti and Zr), 5 (V, Nb, and Ta), and 6 (Mo and W), the *D(M–O)* values are maintained approximately at the same level, and are considerably higher than those in groups 10 and 11, but moving down these groups the TriboEE intensity progressively increases. (2) When the metals in groups 4, 5, and 6 are rearranged in periods 4, 5, and 6, it is seen that, moving to the left of periods 4 (Ti and V), 5 (Zr, Nb, and Mo), and 6 (Ta and W), the TriboEE intensity increases with the increase in the *D(M–O)* values. This trend is also confirmed in Table 1. (3) In groups 10 (Ni, Pd, and Pt) and 11 (Cu, Ag, and Au), moving down the groups, the *D(M−O)* value slowly decreases, interestingly, while the TriboEE intensity tends to increase. (4) When the metals in groups 10 and 11 are rearranged in periods 4, 5, and 6, it is seen that, moving to the right of periods 4 (Ni and Cu), 5 (Pd and Ag), and 6 (Pt and Au), the TriboEE intensity increases with the decrease in *D(M–O)* values. This trend is also confirmed in Table 1. Previously, it was reported that the increase in the TriboEE intensity for groups 4, 5, and 6 can be attributed to the decrease in the WF or photothreshold (Table 3) [8]. Moving down groups 10 and 11, the increase in TriboEE can be associated with the lowering of the *D(M–O)* values. This is a new finding. From the above facts, I think that, for the metals in groups 4, 5, and 6, the electrons predominantly pass over the top of the surface barrier consisting of a surface oxide layer, while for the metals in groups 10 and 11, the electrons can preferentially tunnel through the surface barrier because of the decrease in the thickness of the surface barrier, as explained later.

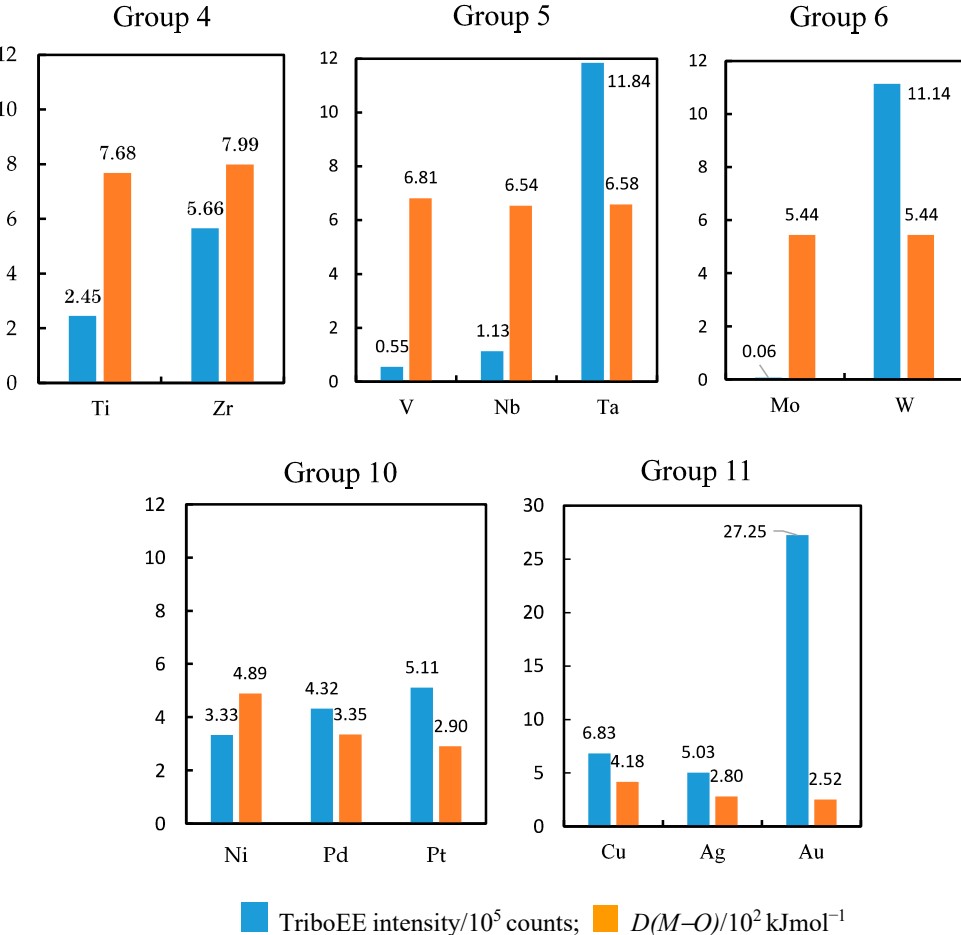

**Figure 9.** The relationship between the TriboEE intensity and the *D(M–O)* values for metals in groups 4, 5, 6, 10, and 11.

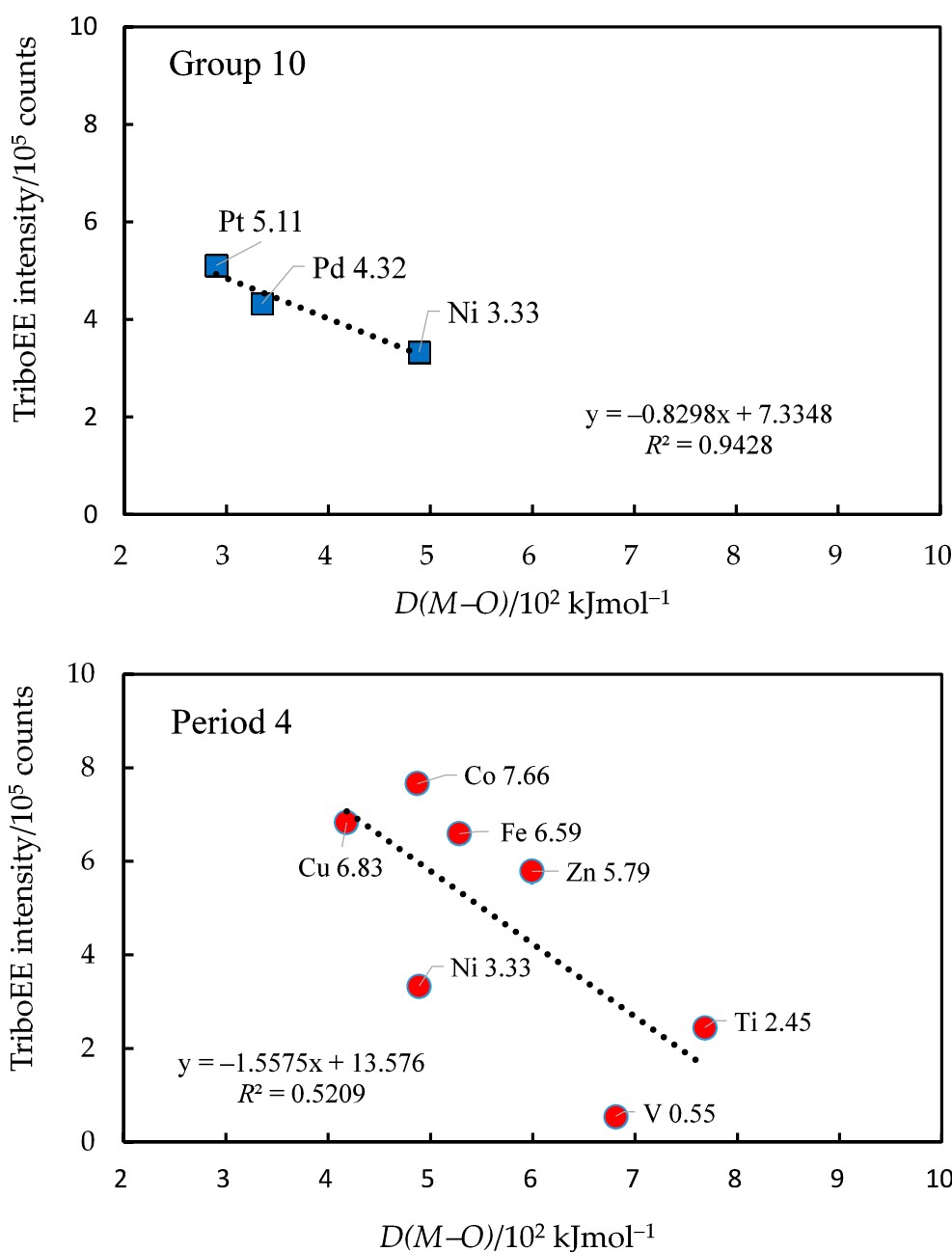

**Figure 10.** The plots of the TriboEE intensity against *D(M–O)* values for metals in group 10 and period 4.

**Table 3.** The work function [1], photothreshold at 298K [2], and electrical conductivities at 295K [3] of the metals used in the periodic table.

| Group | 4 | - | 5 | - | 6 | 8 | - | 9 | - | 10 | - | 11 | - | 12 | 13 | 14 |
|---|---|---|---|---|---|---|---|---|---|---|---|---|---|---|---|---|
| **Element (Period 3)** | - | - | - | - | - | - | - | - | - | - | - | - | - | - | Al | - |
| Work function (eV) | - | - | - | - | - | - | - | - | - | - | - | - | - | - | 4.28 | - |
| Photothreshold (eV) | - | - | - | - | - | - | - | - | - | - | - | - | - | - | 4.25 | - |
| Electrical conductivity ($10^7$ Sm$^{-1}$) | - | - | - | - | - | - | - | - | - | - | - | - | - | - | 3.65 | - |
| **Element (Period 4)** | Ti | - | V | - | - | Fe | - | Co | - | Ni | - | Cu | - | Zn | - | - |
| Work function (eV) | 4.33 | - | 4.3 | - | - | 4.5 | - | 5.0 | - | 5.15 | - | 4.65 | - | 4.33 | - | - |
| Photothreshold (eV) | 4.59 | - | - | - | - | 5.13 | - | 5.01 | - | 5.10 | - | 4.54 | - | 4.67 | - | - |
| Electrical conductivity ($10^7$ Sm$^{-1}$) | 0.23 | < | 0.5 | - | - | 1.02 | < | 1.72 | > | 1.43 | < | 5.88 | > | 1.69 | - | - |
| **Element (Period 5)** | Zr | - | Nb | - | Mo | - | - | - | - | Pd | - | Ag | - | - | - | Sn |
| Work function (eV) | 4.05 | - | 4.3 | - | 4.6 | - | - | - | - | 5.12 | - | 4.26 | - | - | - | 4.42 |
| Photothreshold (eV) | - | - | 5.10 | - | 4.9 | - | - | - | - | 4.99 | - | 4.28 | - | - | - | 4.74 |
| Electrical conductivity ($10^7$ Sm$^{-1}$) | 0.24 | < | 0.69 | < | 1.89 | - | - | - | - | 0.95 | < | 6.21 | - | - | - | 0.91 |
| **Element (Period 6)** | - | - | Ta | - | W | - | - | - | - | Pt | - | Au | - | - | - | - |
| Work function (eV) | - | - | 4.25 | - | 4.55 | - | - | - | - | 5.65 | - | 5.1 | - | - | - | - |
| Photothreshold (eV) | - | - | 4.58 | - | 4.60 | - | - | - | - | 5.09 | - | 4.56 | - | - | - | - |
| Electrical conductivity ($10^7$ Sm$^{-1}$) | - | - | 0.76 | < | 1.89 | - | - | - | - | 0.96 | < | 4.55 | - | - | - | - |

[1] Michaelson, H.B. *J. Appl. Phys.* 1977, *48*, 4729–33. [2] Momose, Y.; Yamashita, Y.; Honma, M. *Tribol. Lett.* 2008, *29*, 75–84. [3] Kittel, C. *Introduction to Solid State Physics*, 5th ed.; John Wiley & Sons: New York, NY, USA, 1976; pp.170.

Furthermore, Figure 10 shows plots of TriboEE intensity against *D(M–O)* values for the metals in group 10 and period 4. The former was a straight line with a slope of $-0.830 \times 10^3$ counts/kJ·mol$^{-1}$. Interestingly, this is considered to propose a new method to quantitatively estimate the effect of the surface barrier on the TriboEE by the metal-oxygen bond energy of the surface oxide layer. The latter plot for the metals in period 4 shows that, on the whole, the TriboEE intensity tends to increase with decreasing values of *D(M–O)*, although the data points are scattered. I think that, in this case, both EE routes tunneling through the barrier (for Fe, Co, Ni, Cu, and Zn) and passing over the barrier height (for Ti and V) are included, but the former, owing to the lowering of *D(M–O)* values, appears stronger than the latter because of the reduction in the WF or photothreshold, resulting in an increase in TriboEE intensity with decreasing *D(M–O)* values. The relationship between the *D(M–O)* and XPS results and the TriboEE intensity is described in more detail later.

### 3.4. TriboEE Intensity, D(M–O), and Electrical Conductivity of Metals

In the TriboEE measurement, as described in Materials and Methods, a voltage of AV = −94V was applied to the metal sample with respect to the earthed grid of the counter. Regarding the electron source for the TriboEE, we reported that, when metal samples of Fe, Ni, and Cu, insulated by placing a polyethylene sheet of 2 mm thickness between the metal sample and the sample holder, were rubbed with a polyethylene rider, the TriboEE was remarkably weak [7,23]. This proves that the electron transport to the metal sample from the battery used as an AV source becomes crucial in producing the TriboEE. I examined how the electrical conductivity of metals affects the TriboEE. In Table 3, the electrical conductivities of metals are given [24].

Figures 11 and 12 show the relationship between the TriboEE intensity and the electrical conductivity. In Figure 11, with the metals in groups 4 (Ti and Zr) and 5 (V, Nb, and Ta), the TriboEE intensity increases with the increase in electrical conductivity, and in group 6 (Mo and W), the TriboEE intensity increases despite having almost the same value of electrical conductivity, while in groups 10 (Ni, Pd, and Pt) and 11 (Cu, Ag, and Au), the TriboEE intensity tends to increase with decreasing electrical conductivity. The last trend is analogous to that of groups 10 and 11, as shown in Figure 9.

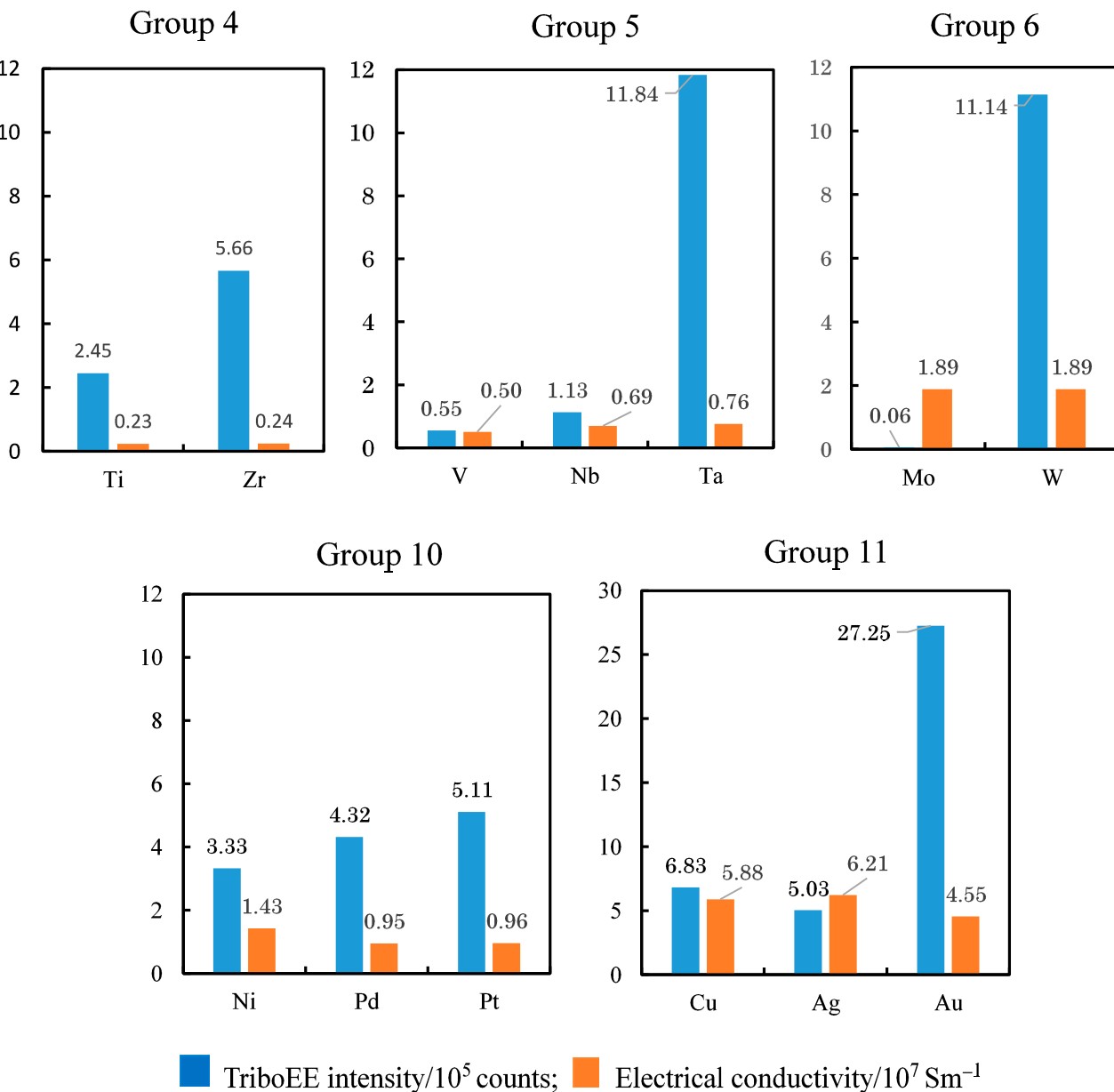

**Figure 11.** The relationship between TriboEE intensity and electrical conductivity values for metals in groups 4, 5, 6, 10, and 11.

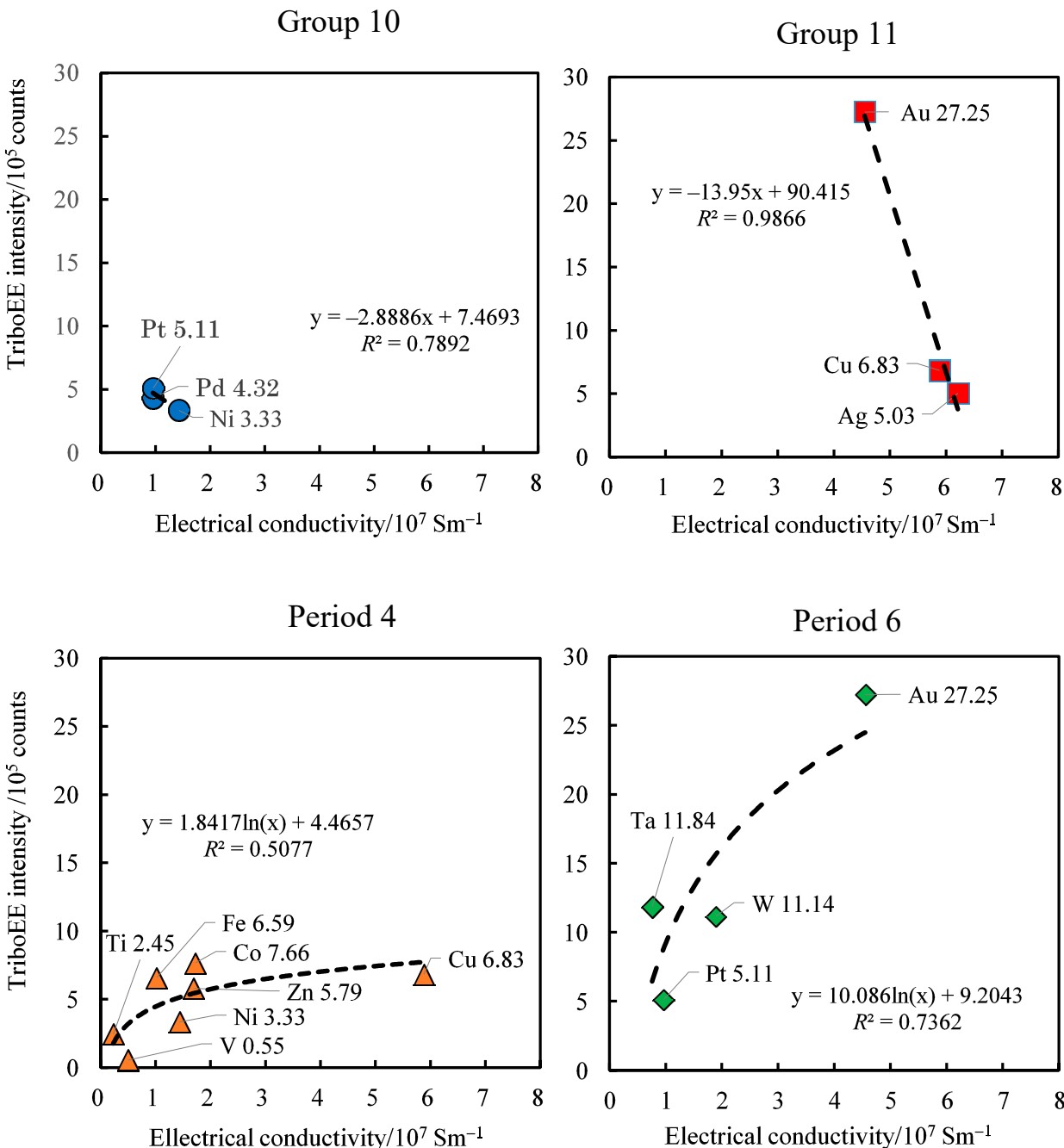

**Figure 12.** The plots of TriboEE intensity against electrical conductivity for metals in groups 10 and 11 (above), and periods 4 and 6 (below).

In Figure 12, we see that, for groups 10 (Ni, Pd, and Pt) and 11 (Cu, Ag, and Au), the plots of TriboEE intensity against electrical conductivity give a straight line with a negative slope. The negative slope ($-13.95 \times 10^{-2}$ counts/Sm$^{-1}$) for group 11 was fivefold greater than that ($-2.89 \times 10^{-2}$ counts/Sm$^{-1}$) of group 10. Figure 12 shows plots of TriboEE intensity against electrical conductivity for the metals in periods 4 and 6. In these cases, on the whole, the TriboEE intensity tends to logarithmically increase with the increasing electrical conductivity. The trend in Figure 12 (below) is the opposite to that observed in Figure 12 (above). Therefore, the effect of electrical conductivity on TriboEE intensity for metals becomes completely different, depending on the metals within groups 10 and 11 or within periods 4 and 6. The reason remains unclear.

Figures 13 and 14 show the relationship between the *D(M–O)* values and the electrical conductivity. This gives the relationship between the data listed in [20–22,24]. Clearly, the metals in groups 4, 5, 6, 10, and 11 that have lower electrical conductivity have greater *D(M–O)* values (Figure 13). Figure 14 shows plots of *D(M–O)* values against electrical conductivity for the metals in periods 4, 5, and 6 and, furthermore, for all metals used. The *D(M–O)* values tend to rapidly decrease, becoming nearly constant with increasing electrical conductivity: *D(M–O)* is proportional to $x^{-n}$, where x is the electrical conductivity and n is 0.2 to 0.3, depending on the period. This suggests that the metal–oxygen bond energy may be reduced by an increase in the electrical conductivity of metals, although the mechanism remains unclear.

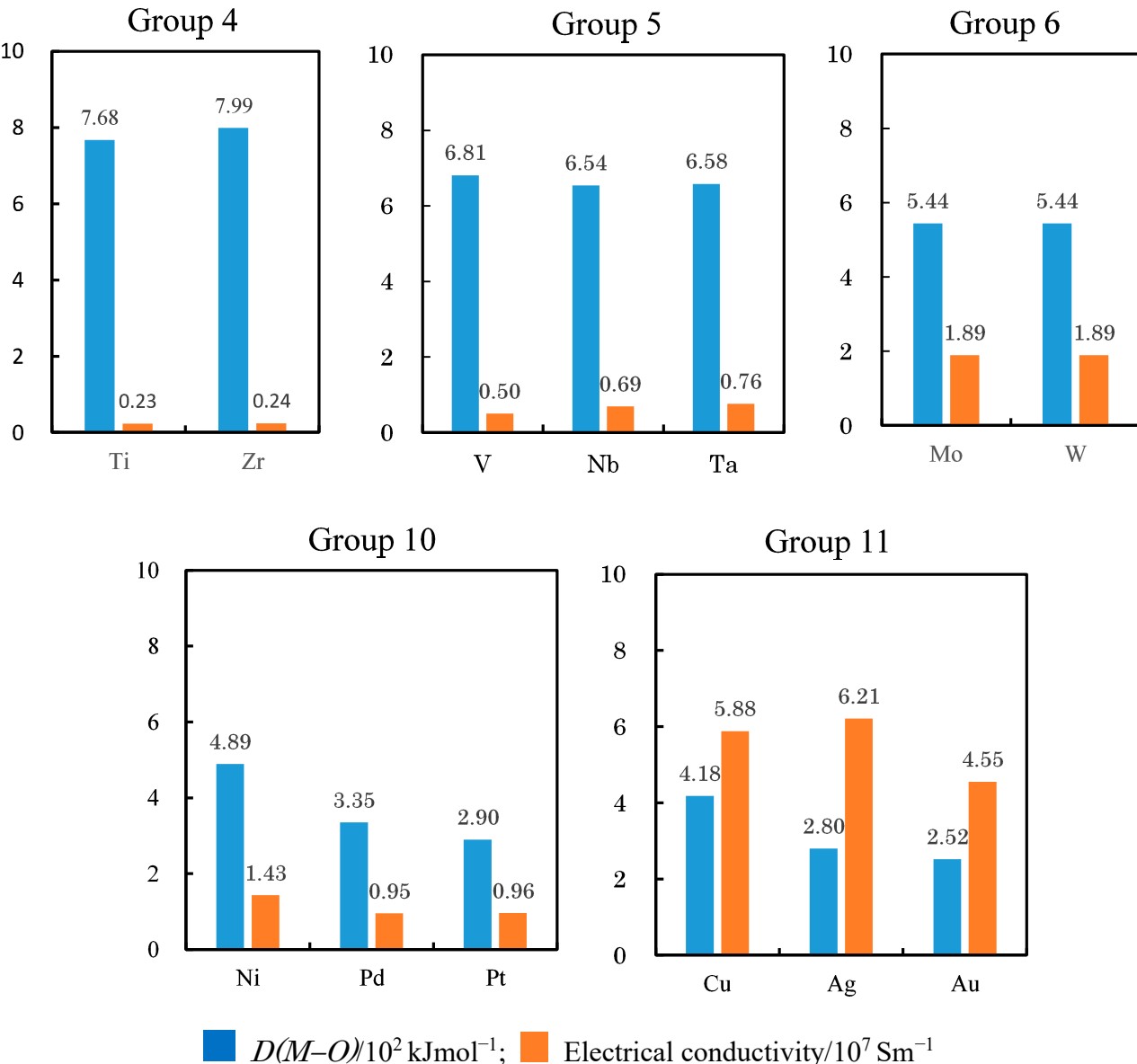

**Figure 13.** The relationship between *D(M–O)* and electrical conductivity values for metals in groups 4, 5, 6, 10, and 11.

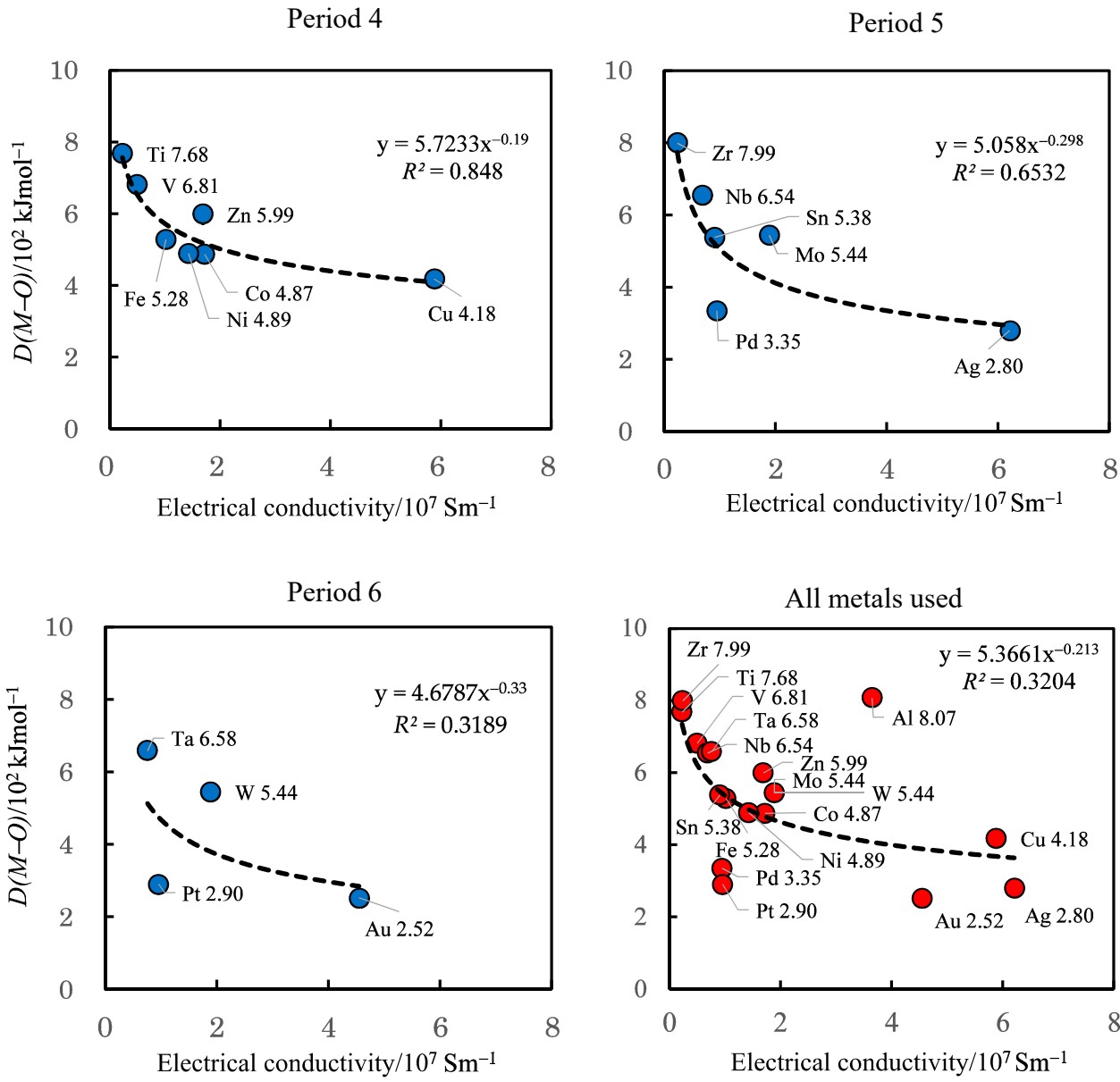

**Figure 14.** The plots of *D(M–O)* values against electrical conductivities for metals in periods 4, 5, and 6, and all metals used.

### 3.5. D(M–O) and XPS Intensity after TriboEE Measurement

We already reported the dependence of the intensities of the XPS spectra on the metals in the periodic groups [7] (Figure 8). Table 4 lists the XPS intensities of F1s, O1s, C1s, and metal core, and the intensity ratio of O1s/Meal core after TriboEE measurement. Figures 15 and 16 show the relationship between the XPS intensities of F1s and O1s after TriboEE measurement (Table 4) and the *D(M–O)* values (Table 1). In Figure 15, in groups 4, 5, and 6, it is seen that, although the *D(M–O)* values remain at nearly the same levels, the F1s and O1s intensities differ depending on the metals, while moving down groups 10 and 11, as the *D(M–O)* values decrease, the F1s intensity increases, while the O1s intensity decreases. This suggests that the decrease in O1s intensity corresponds well with that in *D(M–O)* values, leading to an increase in TriboEE intensity.

**Table 4.** The X-ray photoelectron spectroscopy (XPS) intensities [1] of F1s, O1s, C1s, and metal core spectra and the ratios of O1s/metal core after TriboEE measurement of the metals used.

| Group | 4 | - | 5 | - | 6 | 8 | - | 9 | - | 10 | - | 11 | - | 12 | 13 | 14 |
|---|---|---|---|---|---|---|---|---|---|---|---|---|---|---|---|---|
| Element (Period 3) | - | - | - | - | - | - | - | - | - | - | - | - | - | - | Al | - |
| F1s ($10^3$ counts) | - | - | - | - | - | - | - | - | - | - | - | - | - | - | 8.12 | - |
| O1s ($10^3$ counts) | - | - | - | - | - | - | - | - | - | - | - | - | - | - | 18.97 | - |
| C1s ($10^4$ counts) | - | - | - | - | - | - | - | - | - | - | - | - | - | - | 2.73 | - |
| metal 2p ($10^3$ counts) | - | - | - | - | - | - | - | - | - | - | - | - | - | - | 12.81 (Al2p) | - |
| O1s/metal 2p | - | - | - | - | - | - | - | - | - | - | - | - | - | - | 1.48 | - |
| Element (Period 4) | Ti | - | V | - | - | Fe | - | Co | - | Ni | - | Cu | - | Zn | - | - |
| F1s ($10^3$ counts) | 3.86 | < | 9.65 | - | - | 3.33 | > | 2.07 | < | 8.00 | < | 8.86 | > | 7.23 | - | - |
| O1s ($10^3$ counts) | 10.66 | > | 9.62 | - | - | 7.14 | < | 8.17 | < | 10.24 | > | 9.93 | > | 7.45 | - | - |
| C1s ($10^4$ counts) | 1.54 | < | 2.28 | - | - | 2.11 | ≈ | 2.14 | < | 3.38 | > | 2.08 | ≈ | 2.11 | - | - |
| metal 2p ($10^3$ counts) | 2.85 (Ti2p) | < | 3.08 (V2p) | - | - | 1.88 (Fe2p) | > | 1.32 (Co2p) | < | 1.63 (Ni2p) | > | 0.71 (Cu2p) | < | 1.18 (Zn2p) | - | - |
| O1s/metal 2p | 3.74 | > | 3.12 | - | - | 3.80 | < | 6.19 | < | 6.28 | < | 13.99 | > | 6.31 | - | - |
| Element (Period 5) | Zr | - | Nb | - | Mo | - | - | - | - | Pd | - | Ag | - | - | - | Sn |
| F1s ($10^3$ counts) | 7.60 | > | 6.35 | < | 13.40 | - | - | - | - | 9.65 | > | 9.21 | - | - | - | 11.49 |
| O1s ($10^3$ counts) | 7.21 | < | 15.17 | > | 9.90 | - | - | - | - | 5.79 | > | 2.59 | - | - | - | 11.66 |
| C1s ($10^4$ counts) | 1.66 | < | 1.94 | > | 1.33 | - | - | - | - | 1.52 | ≈ | 1.51 | - | - | - | 2.70 |
| metal3p,3d ($10^3$ counts) | 1.48 (Zr3p) | < | 4.80 (Nb3d) | > | 2.50 (Mo3d) | - | - | - | - | 3.21 (Pd3d) | < | 9.39 (Ag3d) | - | - | - | 4.56 (Sn3d) |
| O1s/metal 3p,3d | 4.87 | > | 3.16 | < | 3.96 | - | - | - | - | 1.80 | > | 0.28 | - | - | - | 2.56 |
| Element (Period 6) | - | - | Ta | - | W | - | - | - | - | Pt | - | Au | - | - | - | - |
| F1s ($10^3$ counts) | - | - | 26.91 | > | 2.28 | - | - | - | - | 23.70 | > | 12.58 | - | - | - | - |
| O1s ($10^3$ counts) | - | - | 14.62 | < | 16.93 | - | - | - | - | 3.76 | > | 2.93 | - | - | - | - |
| C1s ($10^4$ counts) | - | - | 1.97 | < | 2.20 | - | - | - | - | 2.52 | > | 1.58 | - | - | - | - |
| metal4d, 4f ($10^3$ counts) | - | - | 1.59 (Ta4d) | < | 3.65 (W4f) | - | - | - | - | 3.92 (Pt4f) | < | 5.23 (Au4f) | - | - | - | - |
| O1s/metal 4d,4f | - | - | 9.19 | > | 4.64 | - | - | - | - | 0.96 | > | 0.56 | - | - | - | - |

[1] The XPS intensities are the total number of electrons counted per atom for the core spectra, obtained by dividing the difference between the maximum and minimum values—meaning background subtraction—counted for the spectra by its sensitivity.

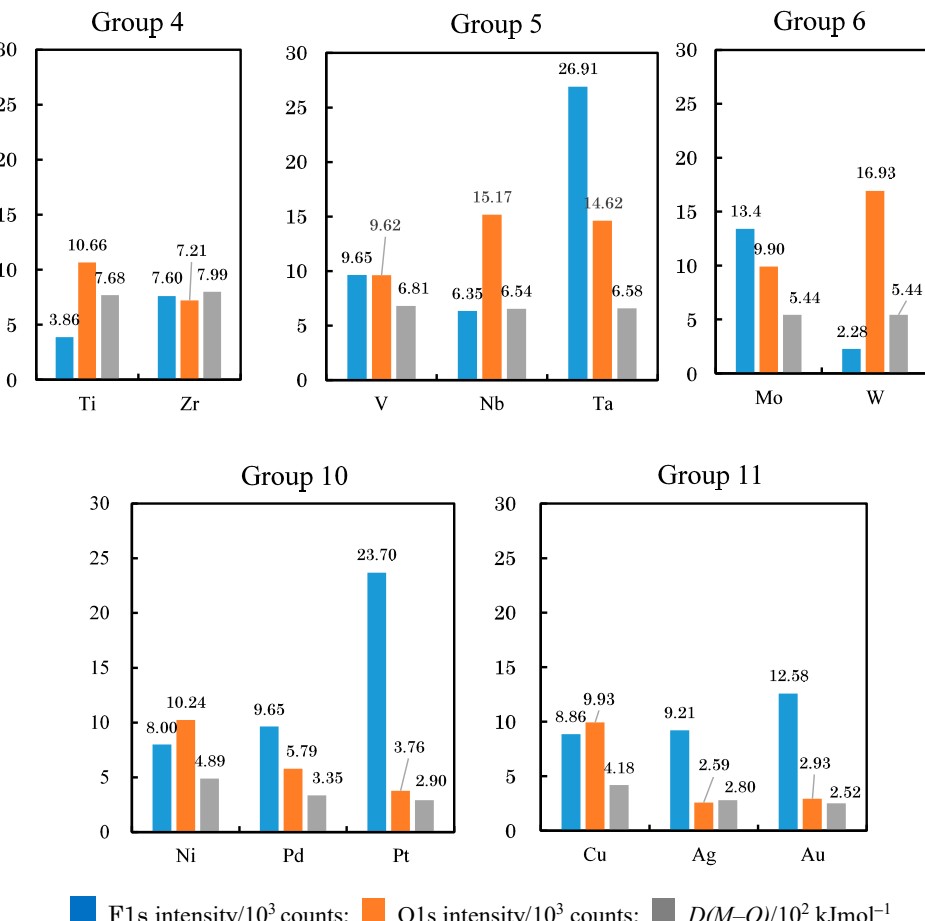

**Figure 15.** The relationship between *D(M–O)* values and XPS intensities of F1s and O1s after TriboEE measurement for metals in groups 4, 5, 6, 10, and 11.

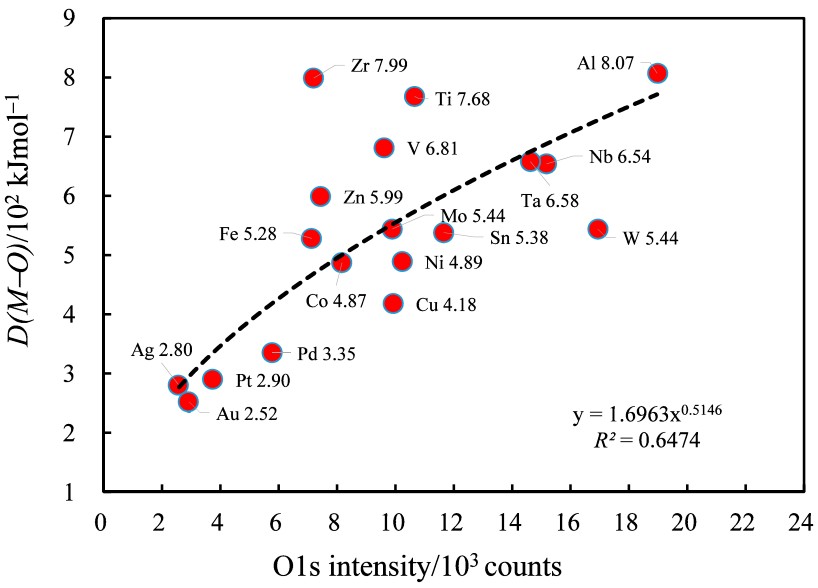

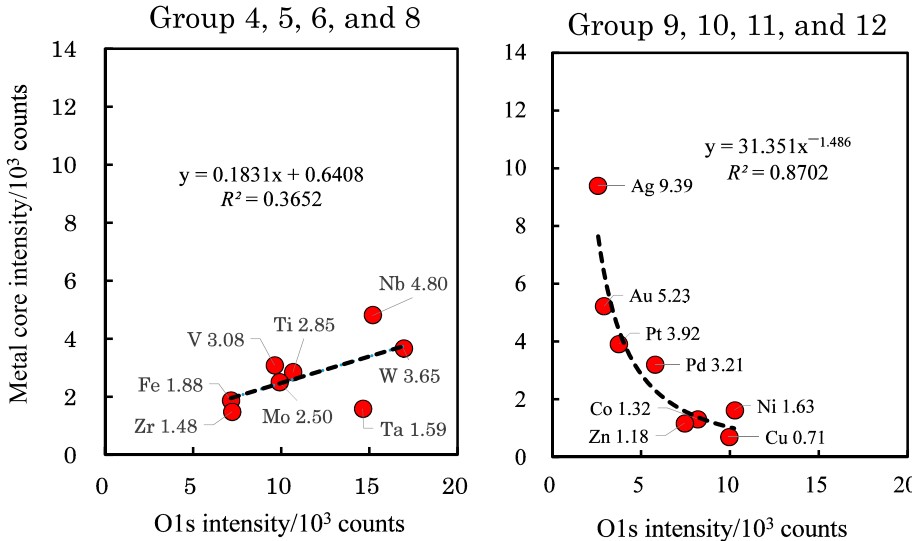

**Figure 16.** The plots of *D(M–O)* values for all metals used (above) and XPS intensities of metal core (below) for metals in groups 4, 5, 6, and 8 and in groups 9, 10, 11, and 12 against O1s intensities after TriboEE measurement.

Regarding the F1s intensity, the following observations are highlighted. (1) The increase in F1s intensity with decreasing *D(M–O)* value suggests that the debris containing fluorine of PTFE rider was strongly adsorbed on the metal surface with a decreased *D(M–O)* value. (2) With Ta and W samples, although the F1s intensities differ greatly (Table 4), their TriboEE intensities are very close (Table 1); therefore, the adsorbed fluorine species seems to have little effect on the TriboEE process. (3) It was, however, reported that a PTFE rider may play the following important role in the TriboEE, explained by two steps. The mechanism is as follows: the first step is electron transfer from a metal substrate to the empty states of a PTFE rider, and the second step is the generation of microplasma by a separation-induced electric field in the gap between the PTFE rider with traps occupied by electrons and the metal surface covered with an oxide film and deposited PTFE debris in the mechanical separation process, finally leading to TriboEE [8,9].

Furthermore, Table 4 indicates that the C1s intensity was much larger than the F1s intensity. It should be noted that the elemental composition of the deposited PTFE was

completely different from the original composition ratio of F/C = 2 and strongly depended on the metals.

In order to make clear in more detail the roles of the surface intensities of O1s and metal core in the TriboEE intensity, I used the intensity ratios of the O1s/metal core. From Table 4, it is seen that the intensity ratio of the O1s/metal core changes as follows: for groups 4, 5, and 6, moving down the groups, the ratio increases as follows: 3.74 (Ti)→4.87 (Zr); 3.12 (V)→3.16 (Nb)→9.19 (Ta); 3.96 (Mo)→4.64 (W). For groups 10 and 11, moving down the groups, the ratio decreases as follows: 6.28 (Ni)→1.80(Pd)→0.96(Pt); 13.99 (Cu)→0.28 (Ag)→0.56 (Au). I described above that, moving down each group, the TriboEE intensity increases (Figure 9 and Table 1). It is seen that the variation of the intensity ratios of the O1s/metal core is in good agreement with the TriboEE intensity. That is, for groups 4, 5, and 6, the oxidization of the surface promotes TriboEE, while for groups 10 and 12, the lowering of the amount of adsorbed oxygen tends to play a predominant role in the increase in TriboEE. Furthermore, it is confirmed that, in the case of paired metals—Ti and V (period 4); Zr and Nb (period 5) except Nb–Mo; Ta and W (period 6)—metals with a higher O1s/metal core ratio give a higher TriboEE intensity. Moreover, with the paired metals Pt and Ag (period 5) and Pt and Au (period 6), except Ni–Cu (period 4), the metal with the lower O1s/metal core yields higher TriboEE intensity. As explained later, an increase in the O1s/metal core ratio may lead to the reduction in the surface barrier height owing to a decrease in WF or photothreshold. On the contrary, the decrease in the O1s/metal core ratio may be related to the reduction in the surface barrier thickness, favorable for tunneling. I think both phenomena may be caused by an increase in the electric field strength applied between the PTFE rider and the metal substrate.

In Figure 16, the dependence of $D(M–O)$ values (all metals) and the metal core intensities (groups 4, 5, 6, and 8, and groups 9, 10, 11, and 12) on the O1s intensities is shown. $D(M–O)$ is considered to indicate the ability of the metal to bind to the oxygen on the metal surface. In Figure 16, on the whole, the $D(M–O)$ value positively increases with increasing O1s intensity. It is seen, however, that there are some irregular data points: the $D(M–O)$ values within groups 4 (Ti and Zr), 5 (V, Nb, and Ta), and 6 (Mo and W) are high and almost the same, respectively; furthermore, moving to the left in period 4 (Ti and V), the $D(M–O)$ value increases with the O1s intensity, while in periods 5 (Zr and Nb) and 6 (Ta and W), the $D(M–O)$ values increase in spite of the decrease in O1s intensity. From these findings, it should be noted that the $D(M–O)$ values of the metals in groups 4, 5, and 6 cannot simply be correlated to the O1s intensity alone. On the other hand, with the metals in periods 4 (Ni and Cu), 5 (Pd and Ag), and 6 (Pt and Au), the $D(M–O)$ values decrease with decreasing O1s intensity in spite of the difference in the oxidation number of the metals. The $D(M–O)$ values of these metals are greatly influenced by the O1s intensity.

In Figure 16, with the metals in groups 4, 5, 6, and 8, which are located on the left side of the periodic table, the metal core intensity tends to slowly increase with increasing O1s intensity, although the data points are scattered; with the metals in groups 9, 10, 11, and 12, which are present on the right side of the periodic table, the metal core intensity rapidly increases with decreasing O1s intensity. Interestingly, these behaviors exhibit a striking contrast; in particular, it is seen that, for Ag, Au, Pt, and Pd, the number of metal atoms bound to one adsorbed oxygen atom is much greater than that of the other metals.

### 3.6. D(M–O) and XPS Intensity before TriboEE Measurement

The XPS spectra before the TriboEE measurement, meaning no sliding with a PTFE rider, are shown in Figures 5–8. In Table 5, the XPS intensities of O1s, C1s, and metal core and the intensity ratios of O1s/metal core data are given.

Figures 17 and 18 show the relationship between the $D(M–O)$ value and the O1s intensity, and the plots of the metal core intensity against the O1s intensity. The behavior is very similar to after the TriboEE measurement, although the metal core and O1s intensities become considerably greater. I think it is useful for metal surface treatments to characterize the metal surfaces covered with a natural oxide layer using the data of XPS and $D(M–O)$.

**Table 5.** The XPS intensities [1] of O1s, C1s, and metal core spectra and the intensity ratio of O1s/metal core before TriboEE measurement of used metals, which were ultrasonically cleaned in a mixture of acetone and petroleum.

| Group | 4 | - | 5 | - | 6 | 8 | - | 9 | - | 10 | - | 11 | - | 12 | 13 | 14 |
|---|---|---|---|---|---|---|---|---|---|---|---|---|---|---|---|---|
| Element (Period 3) | - | - | - | - | - | - | - | - | - | - | - | - | - | - | Al | - |
| O1s ($10^3$ counts) | - | - | - | - | - | - | - | - | - | - | - | - | - | - | 18.52 | - |
| C1s ($10^4$ counts) | - | - | - | - | - | - | - | - | - | - | - | - | - | - | 3.68 | - |
| metal 2p ($10^3$ counts) | - | - | - | - | - | - | - | - | - | - | - | - | - | - | 12.63 (Al2p) | - |
| O1s/metal 2p | - | - | - | - | - | - | - | - | - | - | - | - | - | - | 1.47 | - |
| Element (Period 4) | Ti | - | V | - | - | Fe | - | Co | - | Ni | - | Cu | - | Zn | - | - |
| O1s ($10^3$ counts) | 14.55 | < | 17.10 | - | - | 10.10 | > | 9.28 | > | 9.14 | < | 17.24 | > | 10.38 | - | - |
| C1s ($10^4$ counts) | 1.97 | < | 2.94 | - | - | 2.62 | > | 2.43 | < | 3.11 | > | 2.69 | < | 3.54 | - | - |
| metal 2p ($10^3$ counts) | 3.94 (Ti2p) | < | 5.35 (V2p) | - | - | 2.47 (Fe2p) | > | 1.33 (Co2p) | < | 1.42 (Ni2p) | > | 0.96 (Cu2p) | < | 1.37 (Zn2p) | - | - |
| O1s/metal 2p | 3.67 | > | 3.20 | - | - | 4.10 | < | 6.99 | > | 6.45 | < | 17.91 | > | 7.56 | - | - |
| Element (Period 5) | Zr | - | Nb | - | Mo | - | - | - | - | Pd | - | Ag | - | - | - | Sn |
| O1s ($10^3$ counts) | 14.14 | < | 21.97 | > | 13.76 | - | - | - | - | 10.86 | > | 4.21 | - | - | - | 17.45 |
| C1s ($10^4$ counts) | 1.88 | < | 3.07 | > | 2.63 | - | - | - | - | 2.58 | > | 1.87 | - | - | - | 3.02 |
| metal 3p,3d ($10^3$ counts) | 2.95 (Zr3p) | < | 7.19 (Nb3d) | > | 3.62 (Mo3d) | - | - | - | - | 5.27 (Pd3d) | < | 13.02 (Ag3d) | - | - | - | 6.41 (Sn3d) |
| O1s/metal 3p,3d | 4.79 | > | 3.06 | < | 3.80 | - | - | - | - | 2.06 | > | 0.32 | - | - | - | 2.72 |
| Element (Period 6) | - | - | Ta | - | W | - | - | - | - | Pt | - | Au | - | - | - | - |
| O1s ($10^3$ counts) | - | - | 18.28 | > | 16.72 | - | - | - | - | 5.28 | > | 3.97 | - | - | - | - |
| C1s ($10^4$ counts) | - | - | 2.37 | > | 2.27 | - | - | - | - | 2.48 | > | 2.02 | - | - | - | - |
| metal 4d,4f ($10^3$ counts) | - | - | 1.82 (Ta4d) | < | 3.09 (W4f) | - | - | - | - | 4.84 (Pt4f) | < | 11.78 (Au4f) | - | - | - | - |
| O1s/metal 4d,4f | - | - | 10.02 | > | 5.41 | - | - | - | - | 1.09 | > | 0.34 | - | - | - | - |

[1] The definition of the XPS intensities is the same as in Table 4.

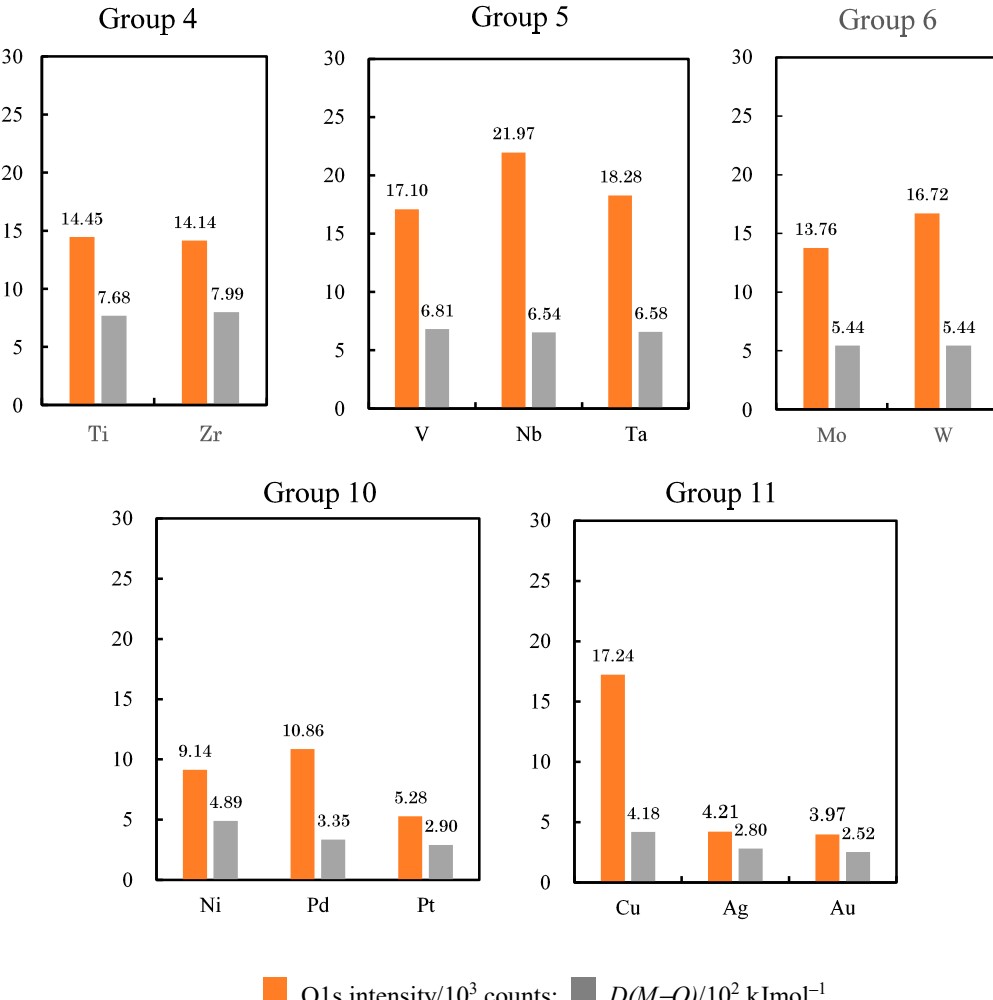

**Figure 17.** The relationship between *D(M–O)* values and XPS intensities of O1s before TriboEE measurement for metals in groups 4, 5, 6, 10, and 11.

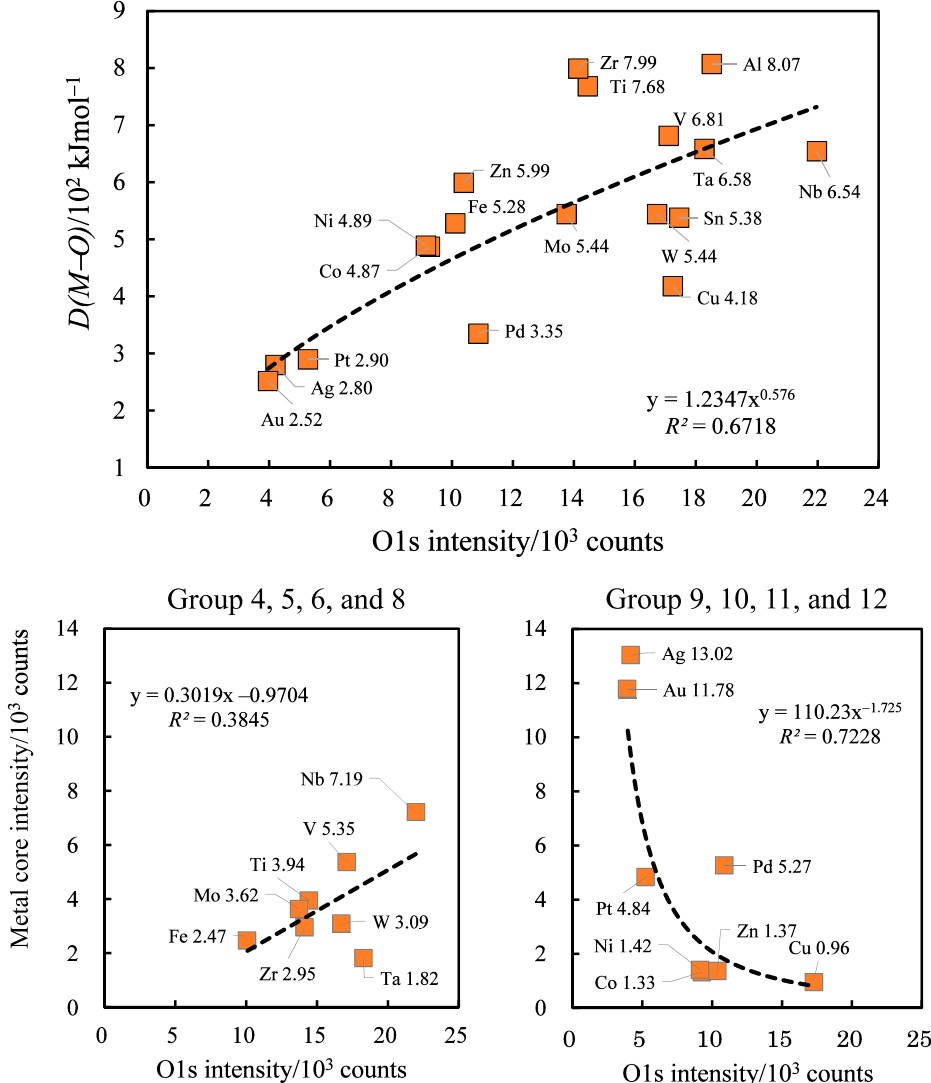

**Figure 18.** The plots of *D(M–O)* values for all metals used (above) and XPS intensities of metal core (below) for metals in groups 4, 5, and 8 and in groups 9, 10, 11, and 12 against O1s intensities before TriboEE measurement.

### 3.7. Scheme of TriboEE by D(M–O) and O1s/Metal Core Ratio

According to [25,26], the models of scheme for the TriboEE using the Schottky effect and tunnel effect are shown in Figure 19a,b, respectively. An electron is confined within a one-dimensional box. The surface oxide layer is represented as a rectangular surface potential barrier of thickness $d_1$ and $d_s$ and height $U_0$. The $U_0$ and $d$ values of the barrier are proposed to include the properties of *D(M–O)* and XPS intensity ratio of O1s/metal core. The values of $d_1$ and $d_s$ are set to correspond to the large and small *D(M–O)* values. The former represents the Schottky effect and the latter the tunnel effect. $\Phi$ is WF or the photothreshold, which is the energy from the Fermi level of the metal to the vacuum level. It is proposed that, in the present experiment, there are two routes in the TriboEE depending on metals: EE passing over the top of the barrier and EE tunneling through the barrier. It should be noted that, according to wave mechanics, so long as the barrier is not infinitely high nor infinitely wide, the electron has a certain probability of passing through the barrier.

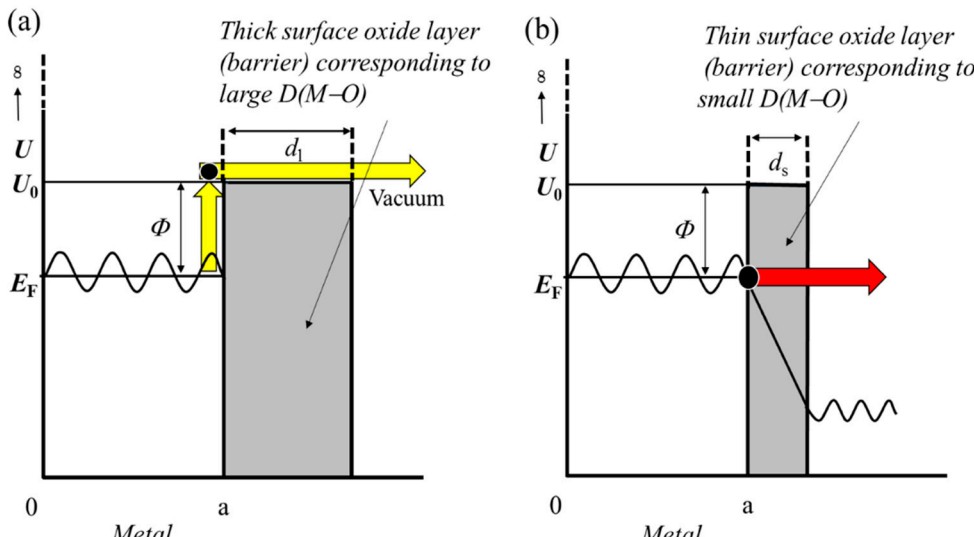

**Figure 19.** Model of scheme for TriboEE from a metal sample covered with surface oxide layer consisting of large (**a**) and small (**b**) *D(M–O)* values during sliding contact with a polytetrafluoroethylene (PTFE) rider, where an external electric field is caused. (**a**) Electrons pass over the top of the barrier (Schottky effect), where the change in work function occurs because of a well-known image force. (**b**) Electrons tunnel through the barrier (tunnel effect).

In Figure 20, the scheme model of the TriboEE in the vicinity at the metal–PTFE interface, which was reported in [9], is rewritten. It consists of two steps: the first step is the electron transfer from a metal substrate to the empty states of a PTFE rider due to an electric field being formed between the metal substrate and the PTFE rider in the sliding contact process; the second step is the generation of microplasma by a separation-induced electric field, formed when the PTFE rider, which possesses traps occupied with electrons from the metal by sliding, is mechanically separated from the metal surface covered with an oxide film and deposited PTFE debris. Finally, this leads to TriboEE. The microplasma generated by frictional electrification was reported by Nakayama and Nevshupa [27].

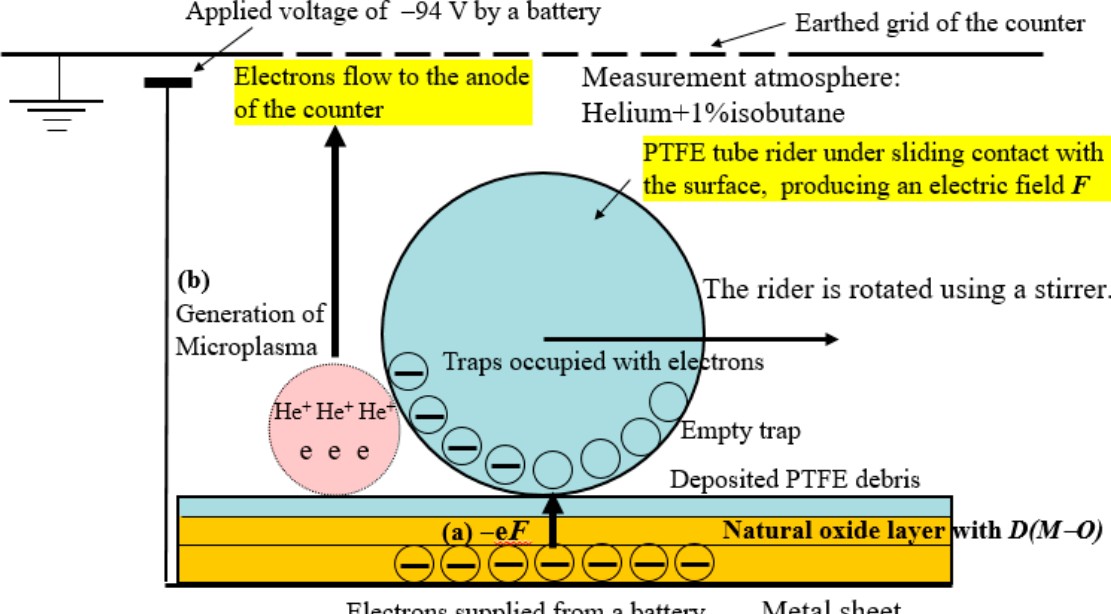

**Figure 20.** The relationship between triboelectron emission from a metal sample covered with a natural oxide layer of *D(M–O)* occurring during sliding process by a PTFE rider and the formation of electric fields: (**a**) a force, $-e\mathbf{F}$, exerted on an electron by an electric field, $\mathbf{F}$, between a metal surface and a PTFE rider; (**b**) flow of electrons from microplasma produced by separation-induced electric field; applied voltage to accumulate electrons.

Regarding the electric fields formed in the TriboEE experiment, three types are considered, as shown in Figure 20a electric field from the metal surface to the PTFE rider with empty traps capable of accepting electrons from the metal in the first step; Figure 20b separation-induced electric field from the PTFE rider with occupied traps to the metal surface in the second step; and the electric field created by applying a negative potential, AV = −94 V, to the metal sample with respect to the earthed grid to accumulate all emitted electrons to the anode of the counter [7]. In the present study, the electric field in the first step is considered. In Figure 19a, the electric field is used for the Schottky effect. This effect is the enhancement of the thermionic emission of a conductor because of the lowering of the work function of the conductor surface, $\Delta\Phi$, represented by Equation (2) and the factor of the increase in the thermionic emission current, $\exp(\Delta\Phi/kT)$, given by the Richardson formula,

$$\Delta\Phi = -e(eF/4\pi\varepsilon_\mathrm{o})^{1/2} \tag{2}$$

where $e$ is the elementary electric charge, $F$ is the electric field strength, $\varepsilon_\mathrm{o}$ is the vacuum permittivity, $k$ is Boltzmann's constant, and $T$ is the temperature in Kelvin. The actual change in work function is relatively small. For example, for $F = 10^3$ V/cm, $\Delta\Phi \approx 0.01$ eV [26]. Therefore, it is suggested that, moving down groups 4, 5, and 6, the increase in intensity ratio of the O1s/metal core given in Table 4 may contribute to the increase in the strength of $F$.

Figure 19b shows a scheme by which the electrons tunnel through the barrier with small $D(M-O)$. When the external electric field becomes on the order of $10^6$ volts per cm, field emission sets in. If the distance $d_\mathrm{s}$ is 1 nm or less, electrons in the vicinity of the Fermi level will tunnel through the barrier [26]. In the Fowler-Nordheim formula known as an elementary field electron emission theory, the emission current ($I$, unit: A/cm$^2$) is represented by Equation (3) using the units of V/cm and eV for the field strength and the work function [28]:

$$I \sim 1.6 \times 10^{-6}\,(F^2/\Phi)\exp\left(-6.9 \times 10^7 \Phi^{3/2}/F\right) \tag{3}$$

This equation suggests that, as the work function of the metal becomes lower and the field strength becomes larger, the current increases. In the present study, with the metals in groups 10 and 11, moving down the groups, reductions in $D(M-O)$ (Figure 9 and Table 1) and the intensity ratio of O1s/metal core (Table 4) take place. I think that these may enhance the accumulation of electrons at the metal-oxide layer interface, leading to an increase in the electric field strength in the form of Equation (3).

Here, the effect of TriboEE during friction should be noted. It was reported that metal sheets abraded with an emery paper were found to act as a strong electrostatic attractive force on polymer sheets charged by triboelectrification, and the force was observed to increase with the decreasing distance between them [14]. This experiment was conducted using a modified electronic balance. Ni and Ti metal sheets and polystyrene and PTFE sheets were used. Therefore, in the present experiment, the metal surface during rubbing is considered to have the action to attract the PTFE rider, probably causing an increase in the friction coefficient.

## 4. Conclusions

The relationship between the ability of EE from real metal surfaces, the metal–oxygen bond, $D(M–O)$, of the natural surface oxide layer, and the electrical conductivity of the metals has been clarified. (a) The EE ability, called TriboEE intensity, was obtained from the total counts of electrons emitted from 18 kinds of metals occurring during sliding with PTFE for rubbing time (60 min). (b) The $D(M–O)$ was estimated from the maximum value of the heat of formation of metal oxides with various oxidation numbers. (c) The TriboEE intensity and $D(M–O)$ for metals were arranged in groups and periods in the periodic table, and the variation of these values between the metals was compared. (d) The values of the electrical conductivity of metals and the O1s and metal core intensities of XPS spectra

obtained after TriboEE measurement were used. (e) With metals in groups 4, 5, and 6, going down the groups, the TriboEE intensity increased in spite of little change in the *D(M–O)*, while with metals in groups 10 and 11, going down the groups, the TriboEE intensity increased with decreasing *D(M–O)*. (f) With metals in periods of the periodic table, the TriboEE intensity increased, tending to become constant with increasing electrical conductivity, while *D(M–O)* decreased and slowly came to a constant level with increasing electrical conductivity. (g) *D(M–O)* values for all used metals tended to increase with increasing O1s intensity. *D(M–O)* values for metals in groups 10 and 11 were limited in the range of a lower O1s intensity. (h) The metal core intensity with metals in groups 9, 10, 11, and 12 rapidly increased with decreasing O1s intensity. (k) It was proposed that the surface oxide layer acts as a surface barrier for the EE, which can be ascribed to electric field caused at the surface oxide layer during the sliding process with PTFE, and the EE takes either the electron transfer passing through the surface oxide layer from metal substrates, called the tunnel effect, or the electron transfer passing over the top of the surface barrier, called the Schottky effect. The metals were divided into two types based on the electron transfer. The metal surfaces with a small amount of oxygen predominantly take the tunnel effect, while those with a large amount of oxygen favor the Schottky effect.

This study suggests what kinds of metals play an important role in the interaction with their environments through electron transfer as well as how the oxygen adsorbed at the metal surfaces can govern the ability of the surfaces to emit electrons. Furthermore, it is suggested that the electrical conductivity of metal wires can be influenced by the formation of oxide layers during sliding against PTFE.

**Funding:** This research received no external funding.

**Institutional Review Board Statement:** Not applicable.

**Informed Consent Statement:** Not applicable.

**Data Availability Statement:** Data is contained within the article.

**Acknowledgments:** The author would like to thank Yusuke Yamashita, a former student of Y.M., for performing the TriboEE and XPS measurements. The author also gratefully acknowledges Keiji Nakayama of Institute of Mesotechnology and Takao Sakurai of Ashikaga Institute of Technology for their continuing encouragement in the study.

**Conflicts of Interest:** The authors declare no conflict of interest.

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
