# Peer review of "Electron Transfer through a Natural Oxide Layer on Real Metal Surfaces Occurring during Sliding with Polytetrafluoroethylene: Dependence on Heat of Formation of Metal Oxides"

_coatings, doi:10.3390/coatings11010109_

Round 1

Reviewer 1 Report

It is a well built paper which incorporates the compulsory problems:  state of the art and why the proposed research was needed. How has been accomplished this target is  clearly presented in the following chapters.

The  manuscript offers enough details in methodology and contains a  large amount of experimental work accompanied by theoretical explanations.

The results are clearly  presented in a sufficient number of diagrams and tables.

In reviewer’s opinion the conclusion chapter  has to point out the main achievements. In this respect, it might be beneficial to reorganize the final conclusion, chapter (4),  with a structure ((a), (b), (c),...).

As single author of the  manuscript it is unusual to start the conclusion chapter with "I reexamined the relationship...". A possible alternative might be: The relationship between....has been reexamined and...".

Author Response

Dear Reviewer 1,

Thank you very much for reviewing my paper.

In lines 560-588, I corrected the conclusions following your suggestions.

Thank you very much in advance,

Yoshihiro Momose

Reviewer 2 Report

The manuscript entitled “Electron transfer through a natural oxide layer on real metal surfaces occurring during sliding with polytetrafluoroethylene: Dependence on heat of formation of metal oxides” has investigated electron emission from real metal surfaces occurring during sliding contact with polytetrafluoroethylene rider. Before the manuscript can be further considered for possible publication, I have several concerns and questions that should be well addressed first.

  1. The tense of the language is inconsistent. For example, line 26 to 28 “The XPS show that……was different”. Line 75 to 76 “ and found that there is”. Line 79 to 80 “It was noticed that these trends are very significant, and often form the basis for catalysis design.” Line 98 “pointed out that it is involved……”.
  2. The content in Table 1 to 5 are poorly displayed, e.g., the figures belonging to one blank is suggested to appear in one line for better visualization.
  3. In the figure captions of Fig.3, “((Cu, Ag, and Au)” should delete one bracket.
  4. Figure 5 to 8 are ambiguous and of poor quality that are hard to see.
  5. Figure 11 loses “Group 4、Group 5 and Group 6.
  6. Some latest works on sliding and surface oxide layer can be cited: Wear 436-437 (2019) 203037; Journal of Alloys and Compounds, 2019, 776: 447-459; Journal of Alloys and Compounds, 2019, 780: 671-679.

Author Response

Dear Reviewer 2

Thank you very much for reviewing my paper.

I checked and corrected the paper following your suggestions.

(1) I corrected the language in lines 26, 76, 79, and 98.

(2) Regarding the tables 1 to 5, it was difficult to make them better visualization because of the size. I attach PDF of the tables.

(3) The bracket was deleted.

(4) All the figures 1 to 20 in the paper were replaced by those when the manuscript was first submitted on 2020-12-21, which were of high quality to see.

(5) Corrected Figure 11 was replaced.

(6) I checked three papers the reviewer suggested to be able to cite, but I could not find the description about sliding and surface oxide layer in their abstracts. So I did not cited them.

Thank you very much in advance,

Yoshihiro Momose

Reviewer 3 Report

The electrical conductivity of metal wires can be influenced by the formation of oxide layers during sliding against Polytetrafluoroethylene (PTFE), the intensity of electron emission depending on rubbing time.
The paper studies the electron transfer through a natural oxide layer on 18 rolled metal sheets from various groups of the chemical periodic table.

The paper is original but uses previously published experimental results of the authors [7, 8].  In my opinion, the quality of the article can be further increased considering the next suggestions:

  1. The authors ought to emphasize in the Introduction section the novelty of the paper in relationship with references [7] and [8]. 
  2. The paper looks rather like a technical report, only the essential data from the extensive tables and Figures should be kept.
  3. Which is the effect of TriboEE during friction? Does this phenomenon influence the friction coefficient or the wear volume? 
  4. On line 78, "moving down" is "moving up".
  5. The caption of Figure 19 is too large. The second half of the caption should be moved to the discussions section.
  6. Try to avoid the use of first-person assertions, especially when expressing conclusions, i.e., replace "I" with "it was".

Author Response

Dear Reviewer 3,

Thank you very much for comments and suggestions.

[Response}

For the sentence given in the beginning, This sentence was employed on lines 617-619.

For (1), The novelty of the paper was described lines 72-76.

For (2), Thank you for your suggestion. In Table 4, the result of the C1s intensity was added on lines 444-447.

For (3), The result of attractive force observed between metal surface and polymer was added on lines 579-585. 

For (4), I could not understand the content of the comment.

For (5), The second half of the caption of Fig. 19 was moved on lines 526-530.

For (6), Thank you very much for your suggestion of replacement of "I" with "it was".

Thank you very much in advance.

Yoshihiro Momose

Round 2

Reviewer 2 Report

After reading the revised manuscript, several minor questions need further explanations and corrections.

  1. From your manuscript, how to calculate the metal-oxygen bond energy need further explanation.
  2. What is the meaning ofI10kcps from your pictures of XPS.
  3. In the Figure 13、15 and 17, the color of “Group4” and “Group5” are different from “Group6、Group10 and Group11”.
  4. At the end of manuscript, line 660 to 665, the picture need change typesetting.

Author Response

Dear Reviewer 2,

Thank you very much for your comments and suggestions.

[Response]

For (1), How to obtain the metal-oxygen bond energy is described lines 276-288.

For (2), The meaning of the signs is described lines 213-215.

For (3), The letters of Group 4 and Group 5 in Figures 13, 15, and 17 were made clear.

For (4), I could not understand the content of the comment.

Thank you very much in advance,

Yoshihiro Momose